# Reviving collapsed plant–pollinator networks from a single species

**Gaurav Baruah** ⬛ *, **Meike J. Wittmann**

Faculty of Biology, Theoretical Biology, University of Bielefeld, Bielefeld, Germany

* gbaruahecoevo@gmail.com

## Abstract

Mutualistic ecological networks can suddenly transition to undesirable states due to small changes in environmental conditions. Recovering from such a collapse can be difficult as restoring the original environmental conditions may be infeasible. Additionally, such networks can also exhibit a phenomenon known as hysteresis, whereby the system could exhibit multiple states under the same environmental conditions, implying that ecological networks may not recover. Here, we attempted to revive collapsed mutualistic networks to a high-functioning state from a single species, using concepts from signal propagation theory and an eco-evolutionary model based on network structures of 115 empirical plant–pollinator networks. We found that restoring the environmental conditions rarely aided in recovery of collapsed networks, but a positive relationship between recovering pollinator density and network nestedness emerged, which was qualitatively supported by empirical plant–pollinator restoration data. In contrast, network resurrection from a collapsed state in undesirable environmental conditions where restoration has minimal impacts could be readily achieved by perturbing a single species or a few species that control the response of the dynamical networks. Additionally, nestedness in networks and a moderate amount of trait variation could aid in the revival of networks even in undesirable environmental conditions. Our work suggests that focus should be applied to a few species whose dynamics could be steered to resurrect entire networks from a collapsed state and that network architecture could play a crucial role in reviving collapsed plant–pollinator networks.

## 1. Introduction

When environmental conditions cross certain threshold values, complex ecological communities can exhibit abrupt transitions from a stable desirable state to an undesirable state where the loss of ecosystem functioning and services occurs [1,2]. Examples are shifts of shallow lakes from clear to turbid state [3], the collapse of fisheries [4], or the collapse of vegetation leading to a desertification [5]. Mutualistic networks governed by positive feedback loops are especially vulnerable to changes in environmental conditions [6–8]. Loss or shifts to undesirable states in such complex networks would lead to a loss in ecosystem functions and services provided by them. Thus, we need to understand the occurrence of such tipping points with an aim to reverse such undesirable effects.

**Data Availability Statement:** Empirical networks were derived from public domain resources of https://www.web-of-life.es/map.php?type=5 and clicking download on the webpage. The restoration data of plant-pollinator communities of Seychelles was also collated from pubic database of http://

www.ecologia.ib.usp.br/iwdb/html/kaiser-bunbury_et_al_2017.html. Adjacency data of the networks with R scripts for reproducing the figures and simulations are available in the Zenodo repository link: https://doi.org/10.5281/zenodo.13598906.

**Funding:** This work was supported by the Deutsche Forschungsgemeinschaft (DFG) Walter Benjamin grant number BA 7974/1-1 to GB. The funders had no role in study design, data collection and analysis, decision to publish, or preparation of the manuscript.

**Competing interests:** The authors have declared that no competing interests exist.

Shifts to undesirable states, such as collapses, are primarily caused by external environmental drivers crossing critical thresholds. These pressures include habitat loss, pollution, over-harvesting, and climate change, which can lead to rapid population and community declines. Consequently, there is an urgent need to forecast these rapid declines to preserve biodiversity and the ecosystem functions associated with these ecological communities. Phenomenological statistical tools such as early warning signals have been suggested to help in forecasting such unwanted transitions [9–15]. These early warning signals have been suggested to be useful in predicting population collapses or community-wide transitions as environmental conditions cross a definite threshold. However, there are many drawbacks associated with using such tools as most often these signals forecast transition imperfectly [16–18]. In the event of an abrupt transition, one potential solution is to restore the original environmental conditions where feasible. For instance, restoration of habitats by removing invasive plant species could potentially increase pollinator diversity [8,19]. However, this approach may not always be effective [20–22]. Often, we lack sufficient information or understanding of the system's properties at the brink of collapse. Additionally, ecological systems may exhibit hysteresis, where they can exist in both functional and dysfunctional states under the same environmental conditions, complicating the recovery process [20,23,24]. Consequently, even when previous conditions are restored, the system may never fully recover and could remain in the undesirable state. When a complex ecological community collapses, restoration strategies carry additional risks. For example, attempting to restore or translocate pollinators to revive plant–pollinator interactions may not always be successful, as many pollinators are inefficient at effective pollen transfer, potentially hindering the recovery of these communities [8].

It is crucial to understand how a dynamical ecological network behaves at the brink of a collapse in order to steer it back to its original functionality once such a network collapses. Previous studies have tried to understand how a dynamical system behaves in the vicinity of a transition [25]. However, recovering an ecological network from an undesirable state is another problem that has remained overlooked. To fill this gap, one solution would be to understand what set of structural network properties [25,26] or eco-evolutionary parameters could be pertinent in the reconstruction of a collapsed ecological network. Once this is known, it would then be easier to comprehend the parameter spaces at which an entire network could be resurrected and the steps that would be needed to ensure that.

Two important metrics quantifying the structure of an ecological network are connectance and nestedness [27,28]. Connectance quantifies how well species in an ecological network interact with other species, and nestedness quantifies species interactions of specialists with only species that interact with both generalists and specialists, i.e., how hierarchical the community is in terms of species interactions. These network properties govern how mutualistic networks respond to changes in environmental conditions such as species loss or decrease in interaction strength [25,29–32]. Previous studies have suggested that such network properties are pertinent to maintaining biodiversity and can impact stability, feasibility, and critical transitions [25,29,33]. While such network features are known to be important for maintaining the stability of mutualistic networks, their role in the resurrection of collapsed networks remains under-explored. It is possible that while resurrecting collapsed low-functional networks, some networks never recover to their original functional state, which could be due to the arrangement of species interactions.

In a rapidly changing world, the maintenance of biodiversity is a key target [34]. However, as the human footprint on our planet continues to grow, leading to widespread biodiversity loss and accelerating extinction rates [35,36], forecasting biodiversity collapse should not be the only goal. We thus propose to investigate the revival of collapsed mutualistic networks by utilizing a phenomenon of propagation of perturbation in such ecological networks. For

instance, a previous study by [26] evaluated the spread of a single perturbation on a node over entire networks, finding that such signal spread can be dictated by structural factors of the networks. Since mutualistic ecological networks consist of species interacting in a certain arrangement, and because of positive feedback between species in such networks, perturbation propagation could possibly be used to resurrect networks that have transitioned to an undesirable state. This is akin to a local environmental disturbance that can spread through an entire ecological network and cause local extinctions [27], or a perturbation signal that spreads through components of various interaction networks [26].

Species embedded in complex ecological networks interact with each other through their phenotypic traits [37,38]. For instance, in plant–pollinator networks, successful and beneficial mutualism could be possible through matching phenotypes such as proboscis of pollinators and corolla lengths of plants [39,40]. If species possess similar matching phenotypes, successful interaction could lead to fitness benefits for both interacting species. In such cases, these interacting species could possess intraspecific variation in such phenotypes which could become crucial in maintaining interacting intimacy with multiple interacting species [41]. Such individual variation, although demonstrated to be widespread in empirical studies, has largely been ignored in understanding its dynamical consequences on stability and resilience (but see [12,32,42]). Recently, individual trait variation in species interaction networks has been suggested to have a significant impact on the occurrence of tipping points and abrupt collapses [32]. Incorporating such trait variation in classical phenomenological models could help us understand how networks could respond to environmental perturbation and whether the presence of trait variation could aid in network resurrection when such networks collapse.

In this study, using a dynamical eco-evolutionary framework, we try to revive mutualistic networks from an undesirable alternative stable state to a high-functioning stable state at unfavorable environmental conditions. We investigate how the original architecture of plant–pollinator networks and the presence of trait variation can facilitate the recovery of collapsed networks. Thus the objectives of this study are threefold: first, to evaluate whether complex plant–pollinator networks exhibit hysteresis, thereby affecting their recovery when environmental conditions are restored; second, to determine if these collapsed networks can be revived from a single species even under unfavorable environmental conditions; and third, to assess whether the architecture of plant–pollinator networks and trait variation influence the revival of collapsed networks. We found, while modeling the eco-evolutionary dynamics of 115 empirical plant–pollinator networks, that restoring the original environmental conditions could rarely aid in recovery of collapsed networks due to the presence of hysteresis. But as mutualistic networks do try to recover due to restoration of previous environmental conditions, a positive relationship between network nestedness and pollinator densities emerges. This particular result was qualitatively supported by our analysis of empirical restoration data of plant–pollinator communities from Seychelles [19]. Next, by combining frameworks from signal propagation theory in networks and eco-evolutionary dynamical modeling, we show that network revival instead could be readily achieved by perturbing a single species that controls the response of such dynamical networks, even in environmental conditions where restoration might not lead to any form of recovery. We show that during the resurrection of such collapsed networks at undesirable parameter spaces, levels of trait variation, and eco-evolutionary dynamics could aid in the revival of such networks. Our study indicates that restoring original environmental conditions when possible might rarely lead to the complete recovery of large mutualistic communities, but focus should be instead applied to a few species whose dynamics could steer the entire network to resurrection. In ecological practice, this would relate to increasing the survival of a particular species, or by adding individuals, or by controlling the density of a species constantly for a period of time.

## 2. Methods and models

### 2.1 Modeling framework

We employ an eco-evolutionary modeling framework combined with concepts from signal propagation theory to understand how structural ecological network properties and eco-evolutionary factors such as heritable trait variation could aid in the resurrection of collapsed networks. We build this framework with mutualistic plant–pollinator networks because mutualistic networks have positive feedback loops that result in the emergence of characteristic alternative stable states [43–45]. Signal propagation patterns in a network and their relationship with network topology have been studied in terms of how the system responds to a local perturbation on a node of a network [26]. In this particular case, a local component of the network is regularly perturbed by a signal by increasing its activity, and the penetration of the signal to other parts of the network is studied to characterize the network's overall behavior in the dynamical regime [26]. We employ a similar methodology and study the interaction of network topology, evolutionary dynamics, and trait variation in the ecological system's components to understand the ability of ecological networks to recover from an undesirable collapsed state.

Firstly, we extracted 115 empirical plant–pollinator networks from the www.web-of-life.es database. Please refer to https://doi.org/10.5281/zenodo.13598906 for details. We used their interaction data to parameterize the eco-evolutionary model that we describe below. The interaction data was used to establish the structure of species interaction in a community. The system of eco-evolutionary equations that characterize the dynamical behavior of mutualistic plant–pollinator networks (shown here for the plant species $i$ in a network, where superscript $P$ denotes plants) [30,32] can be written as follows:

$$\frac{dN_i^{(P)}}{dt} = N_i^{(P)} \int r^{(P)}(z,t) p_i^{(P)}(z,t) dz, \tag{1}$$

and

$$\frac{du_i^{(P)}}{dt} = h^2 \int \left(z - u_i^{(P)}\right) r^{(P)}(z,t) p_i^{(P)}(z,t) dz, \tag{2}$$

where

$$r^{(P)}(z,t) = b - \sum_{j=1}^{S_P} \alpha_{ij}^{(P)} N_j^{(P)} + \sum_{k=1}^{S_A} \int \frac{A_{ik} N_k^{(A)} \gamma(z,z\prime)}{1 + H\gamma(z,z\prime) N_k^{(A)}} p_k^{(A)}(z\prime,t) dz\prime. \tag{3}$$

Here, Eqs 1 and 2 describe the dynamics of plant species density $N_i^{(P)}$ and phenotypic trait dynamics of plant species $i$. $z$ is the phenotypic trait of an individual and $u_i^{(P)}$ is the mean phenotype of plant species $i$, and $p_k^{(P,A)}(z,t)$ is the phenotypic distribution of plant/animal species $k$ at time $t$ (for details of the derivation refer to S1 Text section 1; the system of equations for the animal group are similar but see S1 Text section 1). $b$ is the intrinsic growth rate which was fixed at 0 for all species, indicating that species in mutualistic networks relied solely on mutualistic benefits for their survival and growth (Table 1 for details). $A$ is the adjacency matrix of 0 or 1 where 0 would mean no interaction between an animal species and plant species and 1 would mean an existing interaction. This adjacency matrix corresponded to the empirical plant–pollinator interaction data that we extracted from the web-of-life database. $S_A$, $S_P$ are the total numbers of animal and plant species in the network; $\alpha_{ij}^{(P,A)}$ is the competition coefficient among plants or among animals, i.e., the effect of competition from species $j$ on species $i$; $H$ is

**Table 1. Parameters and their values with the description used in the modeling framework.**

| Symbol | Description |
| --- | --- |
| $b$ | Intrinsic growth rate which was fixed at 0 for all species indicating obligate mutualism. |
| $\sigma_i^{(A)}, \sigma_i^{(P)}$ | Trait standard deviation of animal or plant species $i$; fixed at 0.005 for low trait variation and 0.02 for high trait variation. |
| $h^2$ | Heritability of traits, fixed at 0.4. |
| $\gamma_0$ | Average mutualistic strength among species that was in the range of 0.5 to 1.5 which was in the collapse range (see Fig 1) |
| $d_i^{(A)}, d_i^{(P)}$ | Degree of animal/plant species $i$ in a network. |
| $H$ | Handling time of species which was fixed at 0.25 for all species. |
| $A_{ik}$ | Adjacency matrix entry for the interaction between plant species $i$ and animal species $k$ that is either 0 or 1. |
| $\alpha_{ij}^{(A)}, \alpha_{ij}^{(P)}$ | Competition matrix entry for animals/plants where intraspecific competition $\alpha_{ii} > \alpha_{ij}. \alpha_{ii}$ was fixed at 1 and $\alpha_{ij}$ was drawn from uniform distribution of $U[0.0001, 0.001]$. |
| $\omega$ | Width of the pairwise mutualistic Gaussian kernel fixed at 0.35. |
| $u_i^{(A)}, u_i^{(P)}$ | Mean phenotypic trait values of species. The initial starting values were randomly sampled from $U[-0.5, 0.5]$ or $U[-0.25, 0.25]$ or $U[-0.75, 0.75]$ (see S32 and S33 Figs for sensitivity to different mean trait values). |
| $N_i^{(A)}, N_i^{(P)}$ | Density of species (pollinators and plants) from the collapsed state and drawn from a random uniform distribution on $U[0, 0.005]$ mimicking a state where all species had density either 0 or very low. |
| $\nu(T)$ | Forcing strength of a species for a duration of time given by $T$. Forcing strength varied from 0 to 1, 0.5 for Fig 2, and varied for Fig 3. Duration $T$ also varied from 100 to 500. |

the handling time and $H > 0$, which means increases in growth rate due to mutualistic plant–pollinator interaction follows a type 2 functional response curve. We have fixed $H$ at 0.25, which is within the range that was observed in plant–pollinator interactions [46] but we do a robustness check for different handling time values in S28 and S29 Fig). This indicates that increases in mutualistic benefits of a species $i$ are not linear but saturate as the interacting species density increases. $\gamma(z, z')$ is the Gaussian mutualistic interaction kernel given as follows:

$$\gamma(z, z') = \frac{\gamma_0}{d_i^{(P,A)}} \exp \frac{-(z - z')^2}{\omega^2},$$

where $\gamma_0$ represents the average interaction strength, and $\omega$ designates the width of the interaction kernel, and $d_i^{(P,A)}$ is the degree of plant or animal species $i$. Here, average interaction strength trades off with the species degree ensuring that species with many interactions do not become overly abundant [25,32]. We also relax the assumption of a symmetric Gaussian interaction kernel and do a robustness check with an asymmetric interaction kernel (see S1 Text, section 3 for details). Finally, $h^2$ captures the broad-sense heritability of the phenotypic trait of plant and animal species. Here, the mean phenotypic trait $u_i^{(A,P)}$ changes over time in response to changes in mutualistic interactions in the network. Species in mutualistic networks could interact via phenotypes such as the proboscis of pollinators and corolla lengths in plant–pollinator systems [39,40]. Species with similar phenotypes will gain strong mutualistic benefits in terms of growth according to the trait-matching model [37]. The above system of equations for plant–pollinator mutualistic dynamics exhibits alternative stable states as average mutualistic strength, $\gamma_0$, is varied. At a certain $\gamma_0$, mutualistic networks shift abruptly from a high biomass state to an alternative low biomass state [32,44,45]. This $\gamma_0$ is central to the mutualistic interaction networks and changes in $\gamma_0$ could be indirectly linked to changes in environmental conditions such as temperature [47]. Changes in temperature could shift average interaction strength to a point where such a critical transition occurs.

## 2.2 Collapse and hysteresis regime of mutualistic networks

In our modeling framework, we assumed interspecific competition to be generally weaker than mutualism. In addition, intraspecific competition was fixed at 1 and much stronger than interspecific competition (i.e., $\alpha_{ii} >> \alpha_{ij}$), which was drawn from a uniform distribution of $U$ [0.0001,0.001] [32,44]. This specific distribution indicates that species competition was not too strong to dictate dynamics in a mutualistic community. These distributions were particularly used such that local dynamics became feasible and stable [32,48]. In addition, the distribution of intra- and interspecific competition strengths were within the distributions empirically measured in plant–pollinator communities [49] or in plant communities only [50]. We fixed heritability, $h^2$, at 0.4. In plant–pollinator empirical studies, heritability of morphological traits of pollinators that are involved in pollination such as the proboscis was as high as 0.75 with other morphological traits ranging from 0.1 to 0.9 [51]. For plant floral morphology, heritability was also very high with a median value of 0.55, which thus was around the value we used in our modeling framework [52,53].

Next, we evaluated the collapse regime of plant–pollinator networks by decreasing the average mutualistic strength $\gamma_0$ from a higher to a lower value sequentially from 5 to 0 in steps of 0.15. This value of $\gamma_0$ could signify the level of degradation in a plant–pollinator community. Here, $\gamma_0$ quantifies the average strength in species interaction. For instance, low values of $\gamma_0$ could occur when there is a rise in environmental temperature leading to species mismatches in phenology, thereby leading to a low number of plant–pollinator interactions [54]; or could signify the level of habitat degradation such that plants and pollinators occur at low density [55,56]; or invasion by exotic plant species that impacts flowering of native plants and plant–pollinator interactions [19]. For each value of $\gamma_0$, we simulate the dynamics until time $10^3$. Initial species density was fixed at 1 for all species in all the networks. Mean trait values were as of Table 1, i.e., drawn from a random uniform distribution of U[-0.5,0.5]. We then estimated total plant and pollinator density from the last 100 time points. The collapse threshold for species was set to 0.05, i.e., species below this density were considered to be collapsed. As the average strength of mutualistic interactions, $\gamma_0$, changed, species collapsed until the entire mutualistic networks transitioned to an undesirable collapse state. Next, we roughly estimated the collapse regime as the parameter space of $\gamma_0$ where species richness remained below 80% of the maximum possible richness for all networks. We then focused our next part of the analysis around this $\gamma_0$ regime.

In order to evaluate the hysteresis regime and the impact of restoration, we gradually increased the average mutualistic strength $\gamma_0$ sequentially from 0 to 5 in steps of 0.15. For each value of $\gamma_0$, we simulated eco-evolutionary dynamics until $10^3$. In contrast to the simulations of the collapse regime, here the initial starting density of each plant and animal species in all the networks was ensured to be below 0.005, i.e., to start with species were either locally extinct or had very low density. This simulation scenario could be similar to a restoration practice where the original conditions were restored. Here, $\gamma_0$ being restored is a proxy of restoring external environmental conditions. Mean trait values were as of Table 1, i.e., drawn from a random uniform distribution of U[-0.5,0.5]. Intraspecific trait variance was kept at moderate levels for all species, i.e., at $\sigma_i = 0.005$. As $\gamma_0$ increases, if a mutualistic network exhibits hysteresis, the path of recovery from a collapsed state to a fully functional state would be different, i.e., networks could remain in the collapsed state even at high average mutualistic strength, $\gamma_0$.

## 2.3 Empirical plant–pollinator restoration data from Seychelles

We re-analyzed a dataset of plant–pollinator networks from Seychelles sampled over 8 consecutive months distributed into two treatment sites: one restored and the other unrestored [19].

Each of the treatment sites had 8 independent plots of approximately 1 ha in size. In the restored sites, all invasive plants were completely removed. In the unrestored sites, invasive plants were kept intact. Following this, [19] showed that the restored sites had higher plant–pollinator interaction diversity, a higher number of interactions, and higher function in comparison to unrestored sites. Using their empirical data set, we were specifically interested in evaluating how network nestedness of plant–pollinator networks was related to the recovery of pollinators in restored and unrestored sites. We wanted to compare the plant–pollinator recovery data from the Seychelles data set with our hysteresis simulations (Section 2.2) and evaluate whether the recovery density of pollinators was impacted by network nestedness. We thus analyzed how the visitation rate of pollinators and mean pollinator visits were related to the nestedness of networks in the restored and unrestored sites. We quantified the nestedness (NODF) [57] of plant–pollinator networks in both the restored and the unrestored sites for each sampling period. Mean pollinator visits were quantified as the average number of visits of all pollinators for each plant–pollinator network for each month that was sampled for the two treatment sites. The data already included visitation rate and we calculated the mean visitation rate of each network for each month that was sampled in the two treatment sites. Mean pollinator visits and mean visitation rate for each network were used as a proxy for network recovery.

## 2.4 Perturbation regime in mutualistic networks

Restoration of the original environmental conditions may not always be possible. Next, we thus evaluated another intervention process that does not rely on restoring original conditions. Thus, we perturb a single species in a mutualistic network for a certain duration. The rest of the species in the mutualistic network remain unperturbed. Before we perturbed a species in a network for a certain duration when the network has transitioned to the alternative collapse state, we first used the system of eco-evolutionary equations to simulate dynamics till a network reached a positive quasi-equilibrium state. To start with, the initial species densities of all species in a network were fixed at 1, and average phenotypic species values, $u_i$, were randomly sampled from a uniform distribution of $U[-0.5, 0.5]$, or $U[-0.25, 0.25]$ or $U[-0.75, 0.75]$ (see S32 Fig). We then simulate the dynamics of the network over a time period of $10^3$ time points for an average mutualistic strength $\gamma_0 = 4$. Note that this $\gamma_0$ value of 4 does not fall in the collapse regime (Fig 1). At the final time point, we take the final mean trait values of species for final simulations of network resurrection. This was done to initiate conditions that demonstrate the mutualistic network to be at a state of some kind of positive eco-evolutionary equilibrium, meaning that species densities are high, and mutualist partners have adapted to each other dictated by the structure of the network.

From our analysis of the collapse regime from Fig 1, we observed that almost all networks transitioned from a high biomass state to the collapse state in the range of $\gamma_0 = [0.5, 1.5]$. Knowing this, we focused our subsequent analysis of network revival around this range of $\gamma_0$ values. Around these $\gamma_0$ values, hysteresis existed, and thus restoring the original environmental conditions did not result in the recovery of mutualistic networks (Fig 1B). We thus evaluated whether forcing a single species in plant–pollinator networks could instead be used to resurrect such collapsed networks. To do that, as a naive approach, we chose the species with the highest degree (be it a plant or pollinator) in each of the 115 empirical plant–pollinator networks we analyzed. We set the initial densities of all species in mutualistic networks to $N_i^{(A)}, N_i^{(P)} < 0.005$, i.e., the initial $N_i$'s were sampled from a random uniform distribution with a range of $U[0, 0.005]$. The initial mean phenotypic values for a network were taken from the quasi-positive equilibrium when $\gamma_0 = 4$. Thus, to evaluate the temporal propagation of a perturbation and resurrection, we introduce a condition on the focal species to be perturbed as

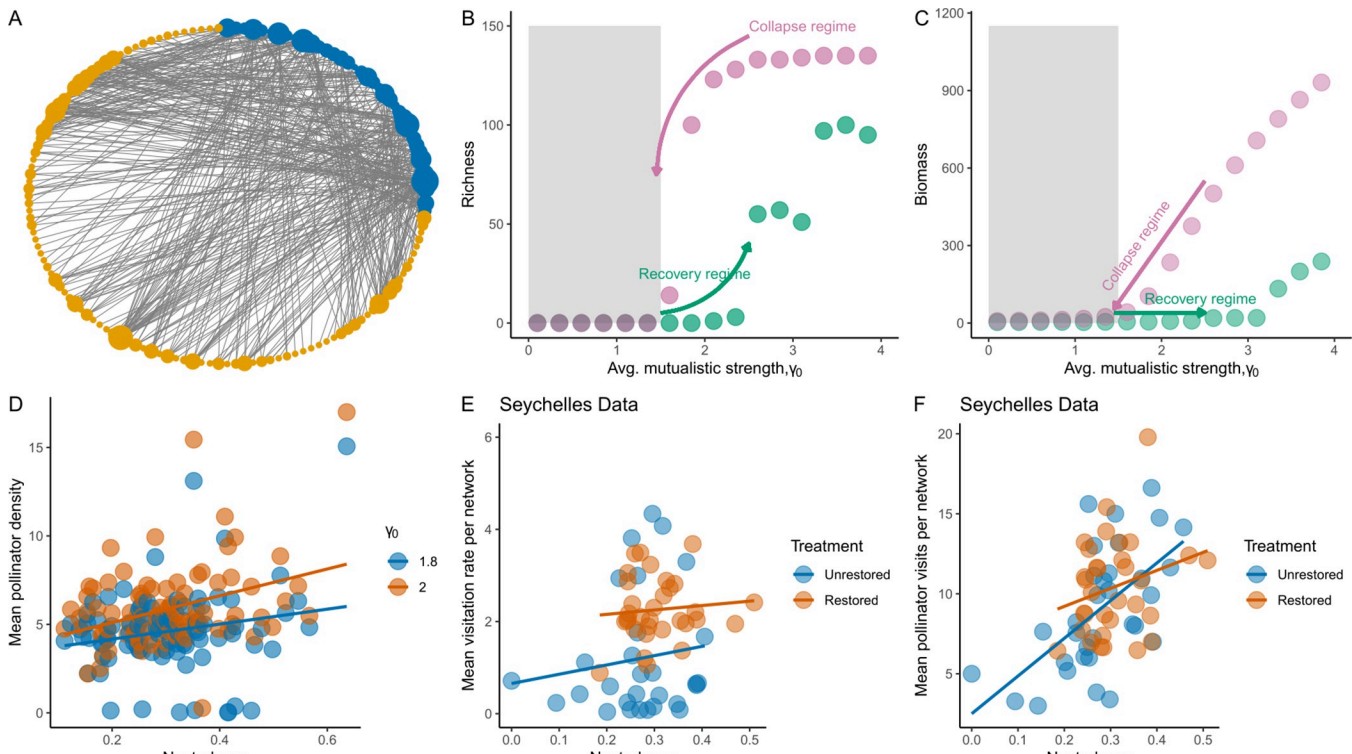

**Fig 1. Collapse regime, hysteresis, and recovery of plant–pollinator mutualistic networks.** (A) An example mutualistic plant–pollinator network of 142 species. The mutualistic network shown here has nestedness of 0.19 and connectance of 0.09. The size of the nodes indicates the number of unique intergroup interactions, i.e., the larger the node size is, the higher the number of interactions. (B, C) As the transition parameter, which is $\gamma_0$, decreases, the mutualistic network transitions from a high-functioning stable state—high species richness in B and high biomass in C—to a low-functioning undesirable state—low richness in B and low biomass depicted by dark pink points. This is the collapse regime, i.e., the mutualistic network completely collapses in the region of, $\gamma_0 = [0,1.5]$. However, as the environmental conditions are restored gradually, i.e., $\gamma_0$ is increased slowly (dark green points), network recovery does not materialize as the system stays at the undesirable low-functioning state (dark green points) and thus exhibits hysteresis. The gray shaded region in B and C represents this collapse regime and the regime where we directed all further analysis. (D) Shown for two levels of $\gamma_0$ outside the collapse regime (gray shaded area in B and C), as the networks recover mean pollinator density at quasi-equilibrium increases as nestedness of networks increases. Each point represents mean pollinator density for a plant–pollinator network, and in total 115 points depicting 115 networks. (E, F) Similar to our simulation results in D, with Seychelles empirical data of network restoration, we observe a similar positive relationship between nestedness and mean pollinator visits (slope = 12.3, s.e. = 4.4, t = 2.7) and mean visitation rate (slope = 8.17, s.e. = 3.8, t-value = 2.126). Underlying data and R scripts for reproducing the figure can be found in https://doi.org/10.5281/zenodo.13598906.

follows:

$$\frac{dN_j^{(A,P)}}{dt} = F_j\left(N_j^{(A,P)}, t\right) + vN_j^{(A,P)}, \tag{4}$$

where, $F_j(N_j^{(A,P)}, t)$ is the right-hand side term of Eq 1 for species $j$, and $v$ is the forcing strength applied for a duration of time $T$. Thus, species $j$ was positively perturbed for a certain duration, and at $t = T$, $v$ was then set to zero. In our simulations, $v$ can range from 0 to 1. This perturbation is akin to increasing the survival or fertility of a species by a certain rate since we fix $b$, the intrinsic growth rate of species to be zero [24]. So any $v$ value greater than 0 would essentially mean a perturbation that increases the growth of a species [24]. To provide a comparison, we also implemented two other forms of perturbation: "targeted perturbation" and "random perturbation." In the targeted perturbation, we selected the 3 species with the highest centrality values for the perturbation regime. In the random perturbation, we randomly chose 1 species to perturb. We also test another specific perturbation that mimics a scenario where we

continuously add some density ($v_C$) of the species for a duration of $T$ time point. In that case (see S6 and S7 Figs for results), Eq 4 would read as follows:

$$\frac{dN_j^{(A,P)}}{dt} = F_j\left(N_j^{(A,P)}, t\right) + v_c.$$

While species $j$ was positively forced, the remaining species remained unperturbed [26]. After the species $j$ is perturbed, we then track the eco-evolutionary dynamics of all the species in the network. Since the parameter regime of $\gamma_0$ where the collapse of mutualistic networks occurred ranged from as low as 0.5 to as high as 1.5, we restricted our perturbation simulations to this parameter range. In other words, for each level of $\gamma_0$ that ranged from 0.5 to 1.5, and for two levels of trait variation (high and low), and for different forcing strength, $v$, we evaluated whether forcing one single species in the collapse regime (i.e., unfavorable environmental conditions) could resurrect networks to a moderate or high-functioning state. We count a species that was not directly perturbed as recovered if the density of the species reached 0.5 at the end of the simulation run from a density of less than 0.005, after the perturbation was stopped. In that way, we track the proportion of species that recovered to a density of 0.5 for each of the 115 plant–pollinator networks. In addition, we also tracked species density after the perturbation was stopped, and the mean and total biomass of networks reached. For two levels of trait variation, high $\sigma_i$ was fixed for all species at 0.02, and low $\sigma_i$ was fixed at 0.005. Here, we evaluated whether moderate levels of phenotypic trait variance in species mean phenotypes could impact the resurrection of collapsed networks. Note that the high level of variance in trait values sampled in this study is at the low end of the spectrum as observed in field studies of plants [58] and pollinators [59,60].

Next, for each network, we estimated common network properties such as connectance and nestedness on the adjacency matrix of networks collected at the initial time point. Connectance was calculated as the total number of interactions divided by the square of the total number of species. For nestedness, we used the commonly used metric known as NODF [57]. Specifically, we asked: Do architectural constraints imposed by how species interact impact the propagation of a perturbation and thereby affect the revival of networks? In asking this question, we also, in an additional S1 Text analyses, tried to investigate the role of indirect effects in recovery biomass of plant–pollinator networks (see S1 Text, section 4 for details). As the number of species in a plant–pollinator network increases, indirect effects will increase [61,62]. Finally, we also tested the impacts of network recovery richness with other network metrics such as modularity [63], mean betweenness centrality, and weighted nestedness [64] (see S1 Text, section 2). We also report the degree distribution of the plant–pollinator networks in S35–S38 Figs.

## 3. Results

We used 115 mutualistic plant–pollinator networks from the web-of-life database (www.web-of-life.es). The networks comprised a total of 7,159 species, with a maximum number of 167 species in a network and a minimum of 8 species. Nestedness (NODF) ranged from as low as 0 to as high as 0.84, while network connectance ranged from a low of 0.035 to a maximum of 0.58.

### 3.1 Collapse of networks and hysteresis

Among 115 mutualistic networks analyzed, the collapse regime, i.e., the parameter region at which the networks transitioned to the collapse state (i.e., less than 90% of its original richness) was $\gamma_0 \leq 1.5$ (Fig 1B and 1C). Thus, $\gamma_0 \leq 1.5$ was considered unfavorable/undesirable

environmental conditions. Out of 115 networks analyzed, 94% of the networks exhibited strong hysteresis, i.e., the networks never recovered even at high values of $\gamma_0 = 4.85$, and complete recovery never occurred despite reverting to the original environmental conditions (Figs 1B, 1C, and S1). Furthermore, we evaluated whether indirect effects could impact hysteresis and found that indirect effects in plant–pollinator networks were mostly negative indicative of the role of competition, which thus led to further disruption in recovery of networks as environmental conditions became better, i.e., $\gamma_0$, increased (S14 and S15 Figs).

## 3.2 Comparison of Seychelles restoration data of plant–pollinator networks and our simulation results

While it is difficult to empirically test all the results of our modeling framework, one testable result of our theoretical study was that as previous environmental conditions, i.e., $\gamma_0$, were restored beyond the unfavorable/undesirable conditions (i.e., $\gamma_0 > 1.5$) network nestedness of plant–pollinator networks had a positive relationship with mean pollinator density (Fig 1D). For Fig 1D, we quantified mean pollinator density at the end of $10^3$ time points for each network (Section 2.2) and nestedness of each of the 115 networks for two levels of $\gamma_0$. Note that nestedness was calculated on the adjacency matrix $A$ in our model simulations as it was fixed for each network we collated from the web-of-life database. In our simulations, the intercept of the positive relationship between mean pollinator density and network nestedness was slightly higher when the average mutualistic strength was higher (Fig 1D). In our framework, we assumed average mutualistic strength, $\gamma_0$, as an indicator of the state of the environment, with high values indicating a better state of the environment. We show that for two specific values of $\gamma_0 = 1.8$ and $\gamma_0 = 2$, and these values fall above the collapse and hysteresis regime, i.e., $\gamma_0 > 1.5$ (Fig 1D). Since in restored sites of Seychelles data, there was higher interaction diversity and higher number of plant–pollinator interactions in comparison to unrestored sites [19], one can qualitatively say that in restored sites the average strength of mutualistic interaction was larger than in unrestored sites. Thus, the restored sites could be qualitatively compared to our modeling scenario of higher average mutualistic strength, $\gamma_0 = 2$, and the unrestored sites could be compared with our modeling scenario of lower average mutualistic strength $\gamma_0 = 1.8$. The results of the analysis of the empirical data showed that as plant–pollinator network nestedness increased the mean number of pollinator visits per network increased (slope = 12.3, s.e. = 4.493, t = 3.712). However, this increase was not significantly different (although the intercept was slightly higher) for networks that were in the restored sites versus those in the unrestored sites (Tables A and B in S1 Text). Similarly, the average visitation rate also had a positive relationship with network nestedness (slope = 8.17, s.e. = 3.8, t-value = 2.126) but there was no significant difference in the relationship between restored and unrestored sites (Fig 1E and 1F) (Tables A and B in S1 Text).

## 3.3 Network resurrection from a single species: Role of trait variation and evolutionary dynamics

When choosing the species with the highest degree for positive perturbation, mutualistic networks could be revived even at a very low mutualistic strength of $\gamma_0 = 1.15$ (Figs 2, 3D, and 3H). Fig 2 shows an example resurrecting dynamics of two mutualistic networks: one with 34 species and another with 49 species. At very low $\gamma_0 = 1.15$, mutualistic networks transition to the collapse state and without any intervention remain at the undesirable collapse state (Figs 1B, 1C, and 2B–2E). With 0.5 forcing strength on the species with the highest degree in both networks for a duration of time ($T = 500$), both the networks easily recovered back to high functionality (Fig 2C and 2F) at low $\gamma_0 = 1.15$. For another large network of 112 species, high

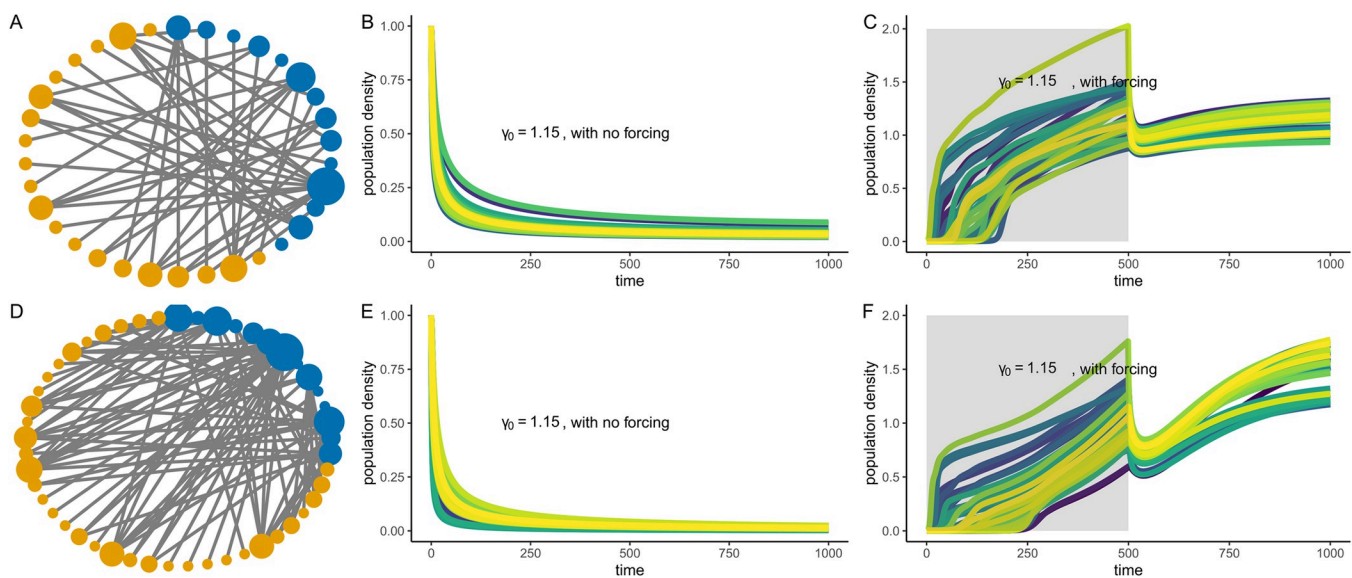

**Fig 2. Network resurrection from perturbing a single species in two contrasting networks of different architecture.** (A) A mutualistic plant–pollinator network with 34 species with nestedness of 0.27 and connectance of 0.15 and (D) a network with 49 species with nestedness of 0.32 and connectance of 0.17. (B, E) Both networks transition to a low-functioning undesirable state at low average mutualistic strength of $\gamma_0 = 1.15$. In B and E, initial species densities were fixed at 1, and trait variation for all species for both the networks was fixed at $\sigma_i = 0.02$. Initial mean trait values $u_i$ were based on Table 1 values. (C, F) Perturbing the species with the highest degree in each of the two networks with a forcing strength of 0.5 at the same average mutualistic strength of $\gamma_0 = 1.15$, for a duration of 500 time points leads to the resurrection of the two networks, i.e., all $N_i > 0.5$. Here for C and F, we considered initial starting species densities to be $N_i < 0.005$ and trait variation for all species to be at a level of $\sigma_i = 0.02$. Underlying data and R scripts for reproducing this figure can be found in https://doi.org/10.5281/zenodo.13598906.

trait variation can aid in resurrecting collapsed networks from a single species perturbation (Fig 3A–3D). With low trait variation, the spread of perturbation from the species with the highest degree played no significant role in resurrecting collapsed networks, i.e., the spread of perturbation was not strong enough to positively impact densities of interacting species (Fig 3E–3H). The presence of evolutionary dynamics also impacted the resurrection of collapsed networks from a single species (in S3–S5 Figs). When evolution was turned off, i.e., heritability was fixed at $h^2 = 0$ for all species, network resurrection from perturbing the most generalist species irrespective of high or low trait variation, failed (S3–S5 Figs). However, this changes the moment evolutionary dynamics come at play, i.e., when heritable variation is no longer zero but $h^2 = 0.4$ for all species, and trait variation was high (S3 and S4 Figs). Although, phenotypic trait change plays an important role, in the absence of evolutionary dynamics, network revival from a single species is a possibility but in a restricted parameter space of the collapse regime (S5 Fig). Network revival from a single species was also slightly impacted by an asymmetric interaction kernel (in S17–S19 Figs).

### 3.4 Network resurrection from a single species: Role of network architecture

As network structure varied, resurrection from a collapsed state to a functional state also varied (Fig 4). Recovery species richness (i.e., all $N_i^{(A,P)} > 0.5$) after perturbing the species with the highest degree was higher when nestedness and connectance of networks were high. Networks that are either small in size, or networks that are high in nestedness or connectance recover to their functional form with only 0.5 forcing strength ($v$) applied to the species with the highest degree (Fig 4A–4C). This was observed more so when species exhibited some amount of trait

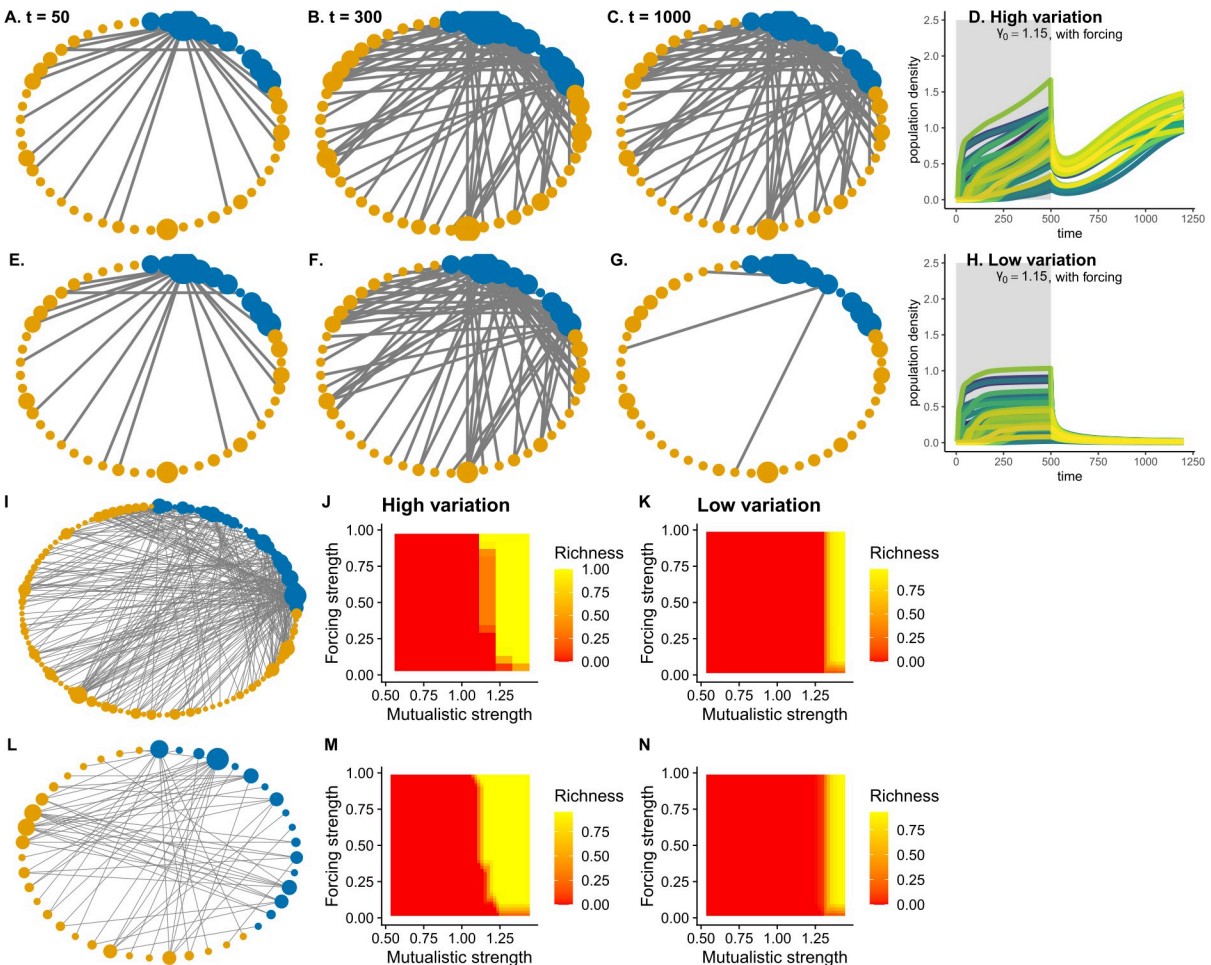

**Fig 3. Role of trait variation for the revival of a network from a single species.** (A–D) High trait variation, i.e., $\sigma_i = 0.02$: A network with 112 species is being reignited from an undesirable stable state, i.e., all species had $N_i < 0.005$ and $\gamma_0 = 1.15$. For resurrecting the collapsed network, a single species with the highest number of interactions was positively perturbed continuously for a duration of 500 time points at a forcing strength of 0.5. (A–C) Time evolution of how the recovery of the network takes place (D) Population dynamics and successful resurrection over time as the species with the highest degree is perturbed. (E–H) Low trait variation case, i.e., $\sigma_i = 0.005$. (E) Initially, at $t = 50$ due to the perturbation that spreads through the network, species recovered their inter-group connections. However, this recovery was temporary as the network fails to recover as shown in (G) $t = 1,000$ and in (H). (I–K) The parameter space for successful resurrection for another network with 112 species differed for the case when species had high trait variation versus when species exhibited low trait variation. For the network with 112 species only at high forcing strength that was greater than 0.25 resurrection of the network was possible at low average mutualistic strength, $\gamma_0 \leq 1.25$. (K) For low trait variation, only at high $\gamma_0 \geq 1.25$, the resurrection was possible. (L–N) For a slightly smaller network with 46 species, we observed similar results. Successful resurrection was possible for high trait variation at low $\gamma_0 < 1.2$ provided there was a small forcing strength. Underlying data and R scripts for reproducing this figure can be found in https://doi.org/10.5281/zenodo.13598906.

variation (Fig 4). Furthermore, the proportion of species that had a density higher than 0.5 at quasi-equilibrium and after species-specific perturbation was stopped, increased as nestedness increased in the unfavorable environmental conditions ($\gamma_0 < 1.5$). This was evident more so when species had higher trait variation in comparison to when species had low trait variation (Fig 5A) for all $\gamma_0 < 1.5$. In addition, the mean species density achieved in each of the 115 networks at unfavorable environmental conditions, i.e., $\gamma_0 < 1.5$ after the perturbation of the most generalist species was stopped was substantially higher when species exhibited high trait variation (Fig 5B). Furthermore, adding a certain density of a species for a certain duration could also be beneficial for reviving collapsed networks (in S6 and S7 Figs). In addition, nonzero intrinsic growth rates impacted network recovery, to the point that having a higher growth

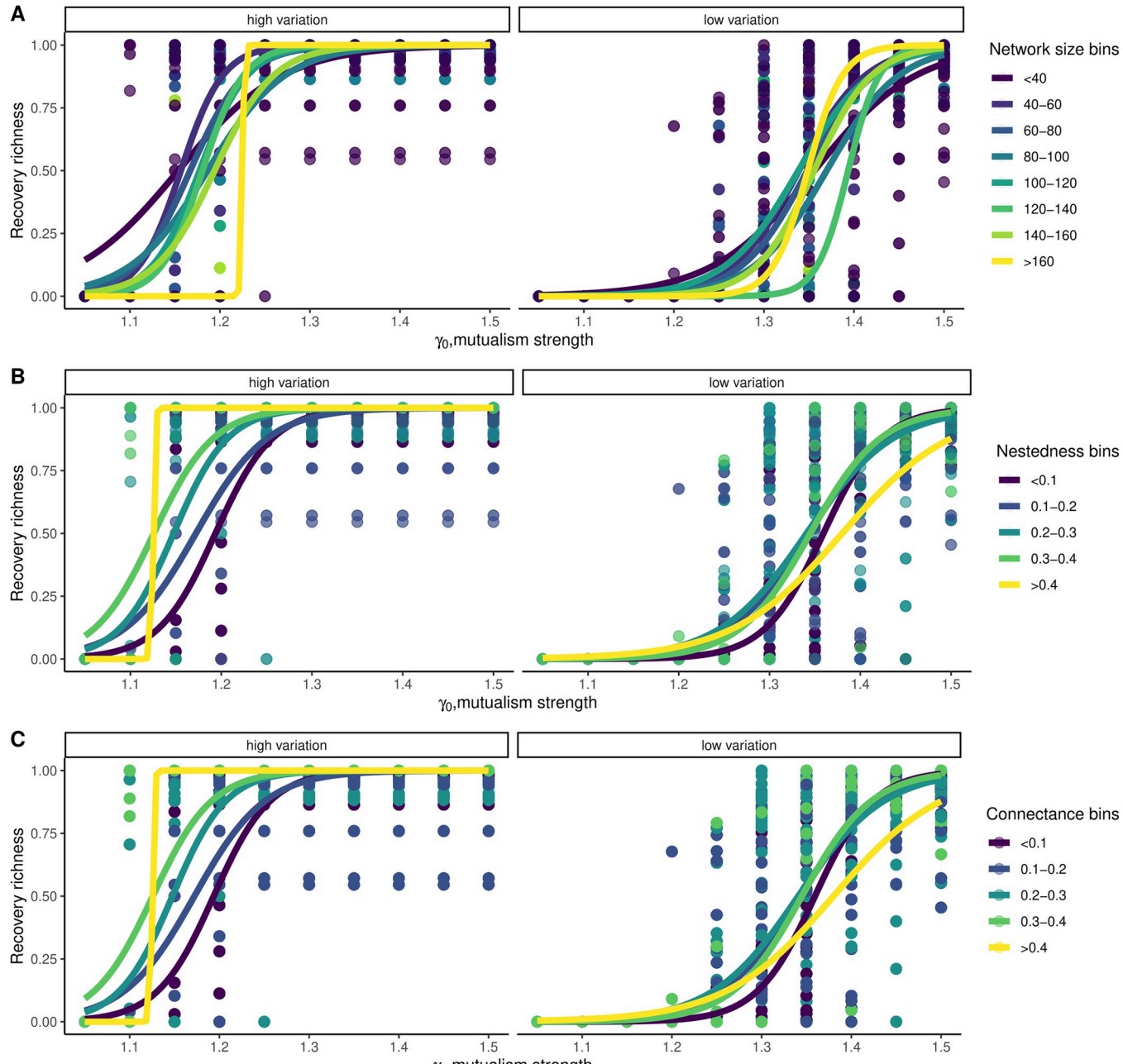

**Fig 4.** Network recovery from perturbing a single species was impacted negatively by network size (A), positively by nestedness (B), and connectance (C), particularly when species had higher trait variation. In each of these networks, only the species with the highest degree, was positively perturbed from a very low density, $N_i < 0.005$, for a duration of 500 time points with a forcing strength of 0.5, while the rest of the species remained unperturbed. Shown here are data from 115 networks. We considered a species to be recovered when $N_i > 0.5$. Recovery richness on Y-axis goes from 0 to 1, 0 indicating no recovery, 1 indicating all species have recovered. $\sigma_i$ was fixed at 0.02 for the high trait variation case and 0.005 for the low trait variation case, respectively. Initial mean trait values were sampled according to parameter values given in Table 1. Underlying data and R scripts for reproducing this figure can be found in https://doi.org/10.5281/zenodo.13598906.

rate allowed better recovery of collapsed networks than others (S30 and S31 Figs). Finally, with higher heritability values (S26 and S27 Figs), lower handling times (S28 and S29 Figs), higher Gaussian kernel width (S24 and S25 Figs), narrower initial mean trait distribution (S32 and

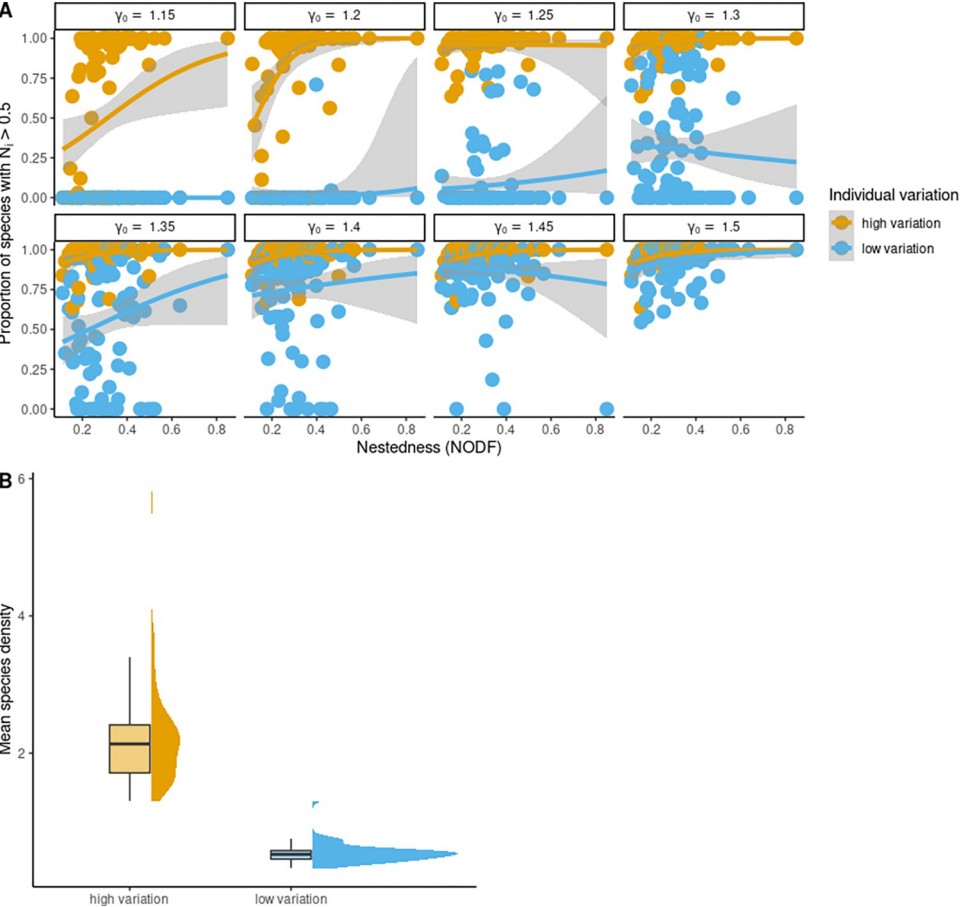

**Fig 5. Impact of plant–pollinator network nestedness and trait variation on plant–pollinator density after perturbation was stopped.** (A) As nestedness increased proportion of species with density greater than 0.5 increased after species-specific perturbation was stopped and more so when species had high trait variation. (B) On average, mean pollinator density across all networks was high when species had high trait variation after recovery for $\gamma_0 > 1$. In (A) to start with, the networks were collated from web-of-life database and species had an initial density of $N_i < 0.005$, and perturbation/forcing strength of 0.5 was applied to the species with the highest degree for a duration of 500 time points. For high trait variation $\sigma_i$ for all species was fixed at 0.02, and for low trait variation $\sigma_i$ was fixed at 0.005. Initial mean trait values were sampled as given in Table 1. Lines with gray shaded regions in A represents confidence intervals of generalized linear model fitting with quasibinomial error distributions. Underlying data and R scripts for reproducing this figure can be found in https://doi.org/10.5281/zenodo.13598906.

S33 Figs), network resurrection from perturbing the most generalist species led to higher recovery richness even at low $\gamma_0$.

## 4. Discussion

We found that complex mutualistic ecological networks exhibited hysteresis and thus never fully recovered as environmental conditions were restored (Figs 1B, 1C, and S1). Thus, restoring the original environmental conditions might not always be a practical solution to resurrect such ecological communities back to high functionality [20]. The end goal of such network resurrection could vary depending on adaptive management strategies or resources available to conduct such a focused resurrection of collapsed communities [65,66]. Nonetheless, it is possible to achieve a highly functional state from a collapsed state, but is concurrently dependent on the level of degradation of the community. The degradation of a community could be due

to the presence of invasive plant species that restrict the growth of native plants and thereby impact plant–pollinator ecosystem functions [19]. The restoration process of removal of invasive plant species or increasing the rate of growth of native plants could indeed improve plant–pollinator functions [19]. Our analysis of empirical data from Seychelles revealed a positive correlation between plant–pollinator functions, measured by pollinator visits and visitation rates, and the nestedness of plant–pollinator networks. Notably, restored communities exhibited a slightly higher intercept in this relationship compared to unrestored communities, suggesting that nested network architecture could facilitate recovery processes. Our modeling results corroborated these findings, showing a positive relationship between mean pollinator density at quasi-equilibrium and network nestedness as the average mutualistic strength, ($\gamma_0$), increased. Increase in ($\gamma_0$) is suggestive of a better environmental conditions (Fig 1D). The key distinction was that nestedness in our models was based on the initial adjacency matrix, whereas in the Seychelles data, it was calculated at each sampling point. In our model, degradation level was indicated by ($\gamma_0$), with lower values signifying higher degradation. In Seychelles, degradation was linked to invasive plants affecting interaction diversity and evenness [19]. Qualitatively, ($\gamma_0 = 1.8$) corresponded to unrestored communities, while ($\gamma_0 = 2$) corresponded to restored sites (Fig 1D–1F).

Restoration of prior environmental conditions when possible could be a way to revive networks to some form of functionality. However, more often than not, it might not be always practically feasible. Thus, one of the challenges was whether it could be possible to revive collapsed networks in unfavorable or undesirable environmental conditions, i.e., $\gamma_0 < 1.5$. Below this range of $\gamma_0$, networks hardly recovered (Figs 1A–1C and S1). Therefore, understanding the dynamic behavior of these networks in these range of $\gamma_0$ values is critical in forecasting their future states in response to perturbation. For ecological networks that have lost more than 80% of their species and/or networks that had species density ranging below a density of 0.005, we used a species-specific perturbation as a tool to steer collapsed networks to a more functional stable state [26]. In a collapsed state, an ecological network has lost its inherent capability to steer itself to some functional form under unfavorable conditions. Thus, outside intervention in some form of positive perturbation could be used to help revive an ecological network [67]. We picked a species that had the highest number of interactions as the focal species to perturb.

When the species with the highest number of interactions was forced positively, mutualistic networks could recover even under unfavorable environmental conditions. This result was dependent on trait variation present, among other factors. We compared two different levels of trait variation. It is to be noted that the two levels of trait variation used were at the lower end of what one could expect when a network is at its undesirable low-functional state [68]. At higher levels of trait variation, when the most generalist species was perturbed, recovery richness after the perturbation was stopped was much higher in comparison to when individual trait variation was lower (Fig 3B–3F). However, this particular result was slightly dependent on the rate of forcing (S6 and S7 Figs) and duration of forcing (S8 Fig).

High trait variation in species would lead to relatively higher trait overlap when compared with low trait variation, which consequently would increase mutualistic benefits [32]. As a result, as the focal generalist species is positively perturbed, this increases its density thereby increasing the overall average mutualistic strength across all the interacting species. The generalist species, thus, acts as a hub to spread the positive perturbation across most species [26], thereby reviving the network from the undesirable state to high functionality in unfavorable environmental conditions. In this particular case, individual trait variation provides not only the base for the spread of perturbation but also the engine for trait dynamics to take over and aid in the resurrection, once the perturbation was stopped (Figs 3A–3C, 5B, S3, and S4). At the

vicinity of a tipping point, a small decrease in $\gamma_0$ could occur too suddenly for the species to adapt and readjust the network accordingly, thereby leading to a sudden collapse. Perturbing a single species for a limited time buys the network time to perform this readjustment. Note also that we have assumed that each species' trait variation remains constant even as the species declines to low density. This may be a reasonable approximation if densities are suddenly reduced. If absolute population sizes are very small, genetic drift would increase, leading to the loss of heritable trait variation, which could then speed up extinction [69]. Even then, we do see that networks do recover at unfavorable conditions even at very low trait variation, but at different parameter ranges. At extremely low mutualistic strength, $\sigma = 0.005$, $\gamma_0 < 1$, large networks with low connectance hardly recover (Fig 4A–4C). Smaller networks with high nestedness, however, do recover even at $\gamma_0 < 1$ (Fig 4A–4C). In principle, the architecture of networks such as nestedness interacts with trait variation to positively impact how species recover even at undesirable environmental conditions after the species-specific perturbation was stopped (Fig 5A). Note that nestedness in such networks correlates with network size and connectance. Thus, changes in nestedness leads to changes in connectance. We conducted additional analyses by fixing network size and connectance and creating plant–pollinator interaction networks with nestedness varying from 1 to 0.38. Our findings confirmed that higher nestedness is still associated with higher recovery species richness, especially when species exhibit high trait variation. Although, this impact is slightly negative when trait variation was low when controlled for network size and connectance (S34 Fig).

When the focal species being positively perturbed was a species with the least amount of interactions or had only a few interactions ($\leq 3$), mutualistic networks did not recover, regardless of the presence of high or low variation, or network topology (S2 Fig). Thus, network resurrection was highly dependent on the type of species that was being perturbed. Our study suggests that, without detailed demographic data, increasing the survival of a super generalist species might be the simplest approach to recovering a collapsed network. However, implementing a strong perturbation on a generalist species poses challenges. If the generalist is a pollinator, it may not be effective in pollen transfer, potentially hindering recovery [8]. Additionally, simply increasing generalist pollinator abundance may not suffice, as they need more than floral resources to thrive [21,70,71]. Conversely, increasing the density of generalist plants could be key, as they support diverse pollinator populations and aid the persistence of rare plants [22,72]. Focusing on a single species is less resource-intensive than targeting multiple species, which involves higher financial and ecological costs. However, targeting multiple species increases the likelihood of successful revival. If a targeted perturbation of one species fails due to unforeseen circumstances, other species may still support the recovery process. Therefore, one effective strategy may be a targeted perturbation, selecting species based on their roles in the community, such as their centrality. In our study, targeted perturbation of the 3 species with the highest centrality in the plant–pollinator network led to faster recovery than the baseline approach of choosing a single species with the most interactions (S20–S23 Figs). Out of the 115 networks analyzed, 106 had plants as the most generalist species. Plants, being immobile, are easier to handle for targeted perturbation compared to pollinators. Therefore, pollinators are not ideal for targeted perturbation. Instead, focusing on increasing the density of generalist plants could be a more effective strategy for recovering collapsed networks [8,73] in addition to restoration of habitats [19,74]. As shown in empirical studies, increasing the density of one or more generalist plant [22,72], or choosing plant species based on high centrality values could be key, as they support diverse pollinator populations and could aid the persistence of rare plants [72]. In comparison to our modeling study, previous studies have indicated that introducing species that could be generalists has positive impact on the structure of plant–pollinator communities [75,76] and further could help in restoring these communities [73].

The knowledge of the system, the type of species interactions, and the presence of feedback loops could be beneficial in managing the recovery of such networks [65]. However, purely competitive communities or communities with predation could potentially constrain the recovery of collapsed networks. For instance, changes in trophic interactions that entail the removal of predators or prey might be important for a successful recovery of vegetation [77,78]. The effectiveness of perturbing the most generalist species to revive a collapsed network thus could vary with the type of species interactions predominant in the community. In complex predator–prey food webs, recovery might not be always guaranteed, especially if only one basal resource species was perturbed [79]. Recovery could depend not only on the number of perturbed basal species but also on the proportion of predation links and the food web structure [79]. Generally, a higher number of predation links could require positively perturbing more basal species for successful recovery from a collapsed state [79]. Communities which harbor negative interactions, such as competitive and predator–prey communities, would thus require different restoration strategies, as such communities have higher number of negative-feedback loops that suppresses the propagation of perturbation [80]. In addition, under harsh environmental conditions, recovery of such communities might be further hampered.

It is to be noted that such proposed revival strategies could have unintended consequences, often failing due to unforeseen factors. For instance, restoring generalist plant numbers might impact soil microbial communities, potentially increasing or decreasing competition with other plants and impacting recovery [81,82]. In addition, one other possible long-term risk would be pollinator rewiring [83]. Increasing generalist plant density could cause pollinators to rewire and specialize on these plants, which in the long term could reduce their interactions with other specialist plants and ultimately could impact ecosystem diversity. Our study does not address these issues, but future research could explore the consequences of adaptive rewiring of pollinators in plant–pollinator during network restoration.

Our modeling framework did not account for species dispersal or spatial structure, both of which can significantly influence revival strategies for collapsed plant–pollinator networks. Dispersal is known to rescue declining populations, enhance productivity, and provide spatial insurance in fragmented landscapes [84,85]. Incorporating dispersal into our model could potentially lead to faster recovery. Additionally, the spatial arrangement of habitat patches could impact recovery, as the structure of habitat patches influences the metacommunity dynamics and processes [85]. One important avenue for future research would be to evaluate how spatial arrangement of habitat patches could interacts with restoration strategies.

We also assumed that trait distributions remain normal, based on our quantitative genetics framework where variance stays constant despite selection on mean traits. In reality, trait distributions could deviate from normality. Relaxing this assumption would require a computationally intensive individual-based approach. Furthermore, we assumed that trait variance is fixed and does not evolve [86]. This assumption may not hold under strong selective pressures to match partner traits. We used low trait variance to mimic low population density conditions. However, if phenotypic variance evolves under strong species-focused perturbation, selection to match multiple partners' traits could increase variance, potentially aiding in faster recovery. Once the network recovers and species traits align, stabilizing selection on the mean could reduce variance. This hypothesis requires further detailed evaluation, particularly on how species' trait variance evolves in response to targeted or species-focused perturbation.

In translating ecological theory to restoration practices in lakes or grasslands, predictive models have included threshold theory in community dynamics [87]. Such predictive models could help in understanding or identifying thresholds [88], which could aid in the recovery trajectory to a desired system [20]. Ecological systems without any sign of hysteresis could be recovered to a desired state when reversing the original environmental driver that caused the

degradation. In the presence of hysteresis, the pathways to collapse and recovery differ markedly as has been demonstrated in a wide variety of ecological systems [23,89,90]. In such cases, restoring the environmental conditions may not result in community recovery and an alternative route to reviving ecological communities would be required. In such systems that are governed by feedback loops, such as the one in this study, i.e., mutualistic networks, our results indicate that focusing on a generalist species through positive perturbation could be beneficial to steer a collapsed network to its original functionality, even at parameter spaces that lead a functional network to collapse, i.e., $\gamma_0 < 1.5$. This idea follows from the fact that such complex communities consist of species that are intertwined in a web of interaction networks, where propagation of perturbation could occur once a particular species is perturbed.

## Supporting information

**S1 Text. Model details and additional results and figures.**
(PDF)

**S1 Fig. Hysteresis regime in mutualistic networks.** (A, B) None of the 115 networks recovered to its original high functional stable state despite reverting to the original environmental conditions. (C) Recovery richness of networks at average mutualistic $\gamma_0 = 4.5$. Here, recovery richness of 1 means all species have recovered, i.e., $N_i^{(A,P)} > 0.5$. Underlying data and R scripts for reproducing this figure can be found in https://doi.org/10.5281/zenodo.13598906.
(TIF)

**S2 Fig.** Recovery richness of networks as a function of Nestedness (top) and connectance (bottom) for a forcing strength $v$ of 0.5 for species that had degrees $d_i \leq 3$. Note that when a specialist species is chosen to be perturbed, i.e., a species with low degree, networks did not resurrect. In the figure the faceted "high" and "low" meant high trait variation and low trait variation. Underlying data and R scripts for reproducing this figure can be found in https://doi.org/10.5281/zenodo.13598906.
(TIF)

**S3 Fig. Impact of evolutionary dynamics and trait variation on network resurrection for a mutualistic network.** (A) A plant–pollinator network of 46 species. (B, C) Population dynamics and evolutionary trait dynamics of the 46 species network in the collapse regime of $\gamma_0 = 1.15$, when trait variation $\sigma_i = 0.005$ was low, and evolution was turned off, i.e., $h_i^2 = 0$ for all species. Here in B and C, at $\gamma_0 = 1.15$ only the most dominant species in terms of interactions was perturbed positively for a duration $T$ of 500 time points with a strength, $v = 0.5$. Network recovery fails here. (D, E) When trait variation $\sigma_i = 0.02$ was high but evolution was still turned off, i.e., $h_i^2 = 0$, network resurrection fails. (F, G) When evolution was turned on i.e., $h_i^2 = 0.4$ but trait variation was low ($\sigma_i = 0.005$), network resurrection still fails. (H, I) Only when trait variation was moderate, i.e., $\sigma_i = 0.02$ and evolution turned on $h_i^2 = 0.4$, network resurrection in the collapse-regime from perturbing the most generalist species succeeds. Here in all the simulations, initial densities of all species $N_i$ were sampled from a uniform random distribution of $U[0, 0.005]$. Forcing strength $v = 0.5$, duration of perturbation $T = 500$. Underlying data and R scripts for reproducing this figure can be found in https://doi.org/10.5281/zenodo.13598906.
(TIF)

**S4 Fig. Impact of evolutionary dynamics and trait variation on network resurrection for another example mutualistic network.** (A) A plant–pollinator network of 31 species. (B, C) Population dynamics and evolutionary trait dynamics of the 31 species network in the collapse

regime of $\gamma_0$ =1.15, when trait variation $\sigma_i$ = 0.005 was low, and evolution was turned off, i.e., $h_i^2 = 0$ for all species. Here in B and C, at $\gamma_0$ = 1.15 only the most dominant species in terms of interactions was perturbed positively for a duration $T$ of 500 time points with a strength, $v$ = 0.5. Network recovery fails here. (D, E) When trait variation $\sigma_i$ = 0.02 was high but evolution was still turned off, i.e., $h_i^2 = 0$. Here, network resurrection fails. (F, G) When evolution was turned on, i.e., $h_i^2 = 0.4$ but trait variation was low ($\sigma_i$ = 0.005), network resurrection still fails. (H, I) Only when trait variation was moderate, i.e., $\sigma_i$ = 0.02 and evolution turned on $h_i^2 = 0.4$, network resurrection in the collapse-regime from perturbing the most generalist species succeeds. Here in all the simulations, initial densities of all species $N_i$ were sampled from a uniform random distribution of $U[0, 0.005]$. Forcing strength $v$ = 0.5, duration of perturbation $T$ = 500. Underlying data and R scripts for reproducing this figure can be found in https://doi. org/10.5281/zenodo.13598906.
(TIF)

**S5 Fig. Comparison of network revival in the presence and absence of evolutionary dynamics on 3 example networks.** (Top) For a network with 49 species: Network revival when forcing the species with the highest number of interactions for a wide range of forcing strength, $v$, and average mutualistic strength for 2 levels of heritability, $h_i^2 = 0$ and $h_i^2 = 0.4$. Even in the absence of heritable variation, i.e., the absence of evolutionary dynamics, network revival from a single species at unfavorable conditions could still be a possibility. (Middle) For a network with 34 species shown for a wide range of forcing strength $v$ and average mutualistic strength and for 2 levels of heritability, $h_i^2 = 0$ and $h_i^2 = 0.4$. (Bottom) For a network with 105 species shown here for a wide range of forcing strength, $v$, and average mutualistic strength and for 2 levels of heritability, $h_i^2 = 0$ and $h_i^2 = 0.4$. Here in all the simulations, initial densities of all species $N_i$ were sampled from a uniform random distribution of $U[0, 0.005]$. Duration of perturbation $T$ = 500, and trait variation was fixed for all species at $\sigma_i$ = 0.02. Underlying data and R scripts for reproducing this figure can be found in https://doi.org/10.5281/zenodo.13598906.
(TIF)

**S6 Fig. Parameter space of network recovery as mutualistic strength and addition of constant density of $v_C$ varies for a 43 species network and perturbation is applied to the species with the highest degree.** Here constant addition of density of 1.2 (forcing on y-axis) would mean that at constant rate a density of 1.2 of the most generalist species is added for a duration of 500 time points. High trait variation: networks recover fully at low $\gamma_0$, whereas for low variation, networks recover only at higher $\gamma_0$ values. Initial species density was below 0.005, mean trait values were sampled from Table 1 in the main text. Low variation $\sigma_i$ = 0.005, and high variation $\sigma_i$ = 0.02. Underlying data and R scripts for reproducing this figure can be found in https://doi.org/10.5281/zenodo.13598906.
(TIF)

**S7 Fig. Parameter space of network recovery as mutualistic strength and constant density forcing strength $v_C$ varies for a 32 species network and perturbation applied to the species with the highest degree.** Here forcing of 1.2 would mean that at consecutive time points a density of 1.2 of the most generalist species is added for a duration of 500 time points. High trait variation: networks recover fully even at low $\gamma_0$, whereas for low variation, networks recover only at higher $\gamma_0$ values. Initial species density was below 0.005, mean trait values were sampled as from Table 1 in the main text. Low variation $\sigma_i$ = 0.005 and high variation $\sigma_i$ = 0.02. Underlying data and R scripts for reproducing this figure can be found in https://doi.org/10. 5281/zenodo.13598906.
(TIF)

**S8 Fig. Parameter space of network recovery as mutualistic strength and forcing duration varies for 3 networks of different structures and perturbation applied to the species with the highest degree.** (A) A network of 61 species, connectance of 0.09, nestedness of 0.19, (D) a network of 101 species with connectance of 0.108 and nestedness of 0.221, and (G) a network of 17 species with a connectance of 0.288, nestedness of 0.292. In (B-E-H) high trait variation: networks recover readily as forcing duration increases even at low levels of $\gamma_0$. (C-F-I) Low trait variation: same networks with low trait variation. Forcing strength was fixed at 0.5, initial species density was below 0.005, mean trait values were sample as from Table 1 in the main text. Low variance $\sigma_i$ = 0.005 and high variance $\sigma_i$ = 0.02. Underlying data and R scripts for reproducing this figure can be found in https://doi.org/10.5281/zenodo.13598906.
(TIF)

**S9 Fig. Parameter space of network recovery as variance in $h^2$ for species and mean $h^2$ is varied for 3 different networks, for a particular mutualistic strength $\gamma_0$ of 1.2, for forcing strength of 0.5 and forcing duration of 500 time points when trait variation was high.** (A) A plant–pollinator network of 35 species. (B) Mean $h^2$ had a larger impact than variance on recovery richness, when the most generalist species is perturbed for a certain duration. (C) A plant–pollinator network of 61 species. (D) A slightly higher $h^2$ leads to full recovery of all species with variance in $h^2$ having very small impact on the outcome. (E, F) Similarly for 101 species network, we observed similar outcomes, with variance in $h^2$ having little impact. Forcing strength was fixed at 0.5, initial species density was below 0.005, mean trait values were sample as from Table 1 in the main text. High variance $\sigma_i$ = 0.02. Underlying data and R scripts for reproducing this figure can be found in https://doi.org/10.5281/zenodo.13598906.
(TIF)

**S10 Fig.** Network recovery from perturbing a single species was impacted moderately by network modularity (A), positively impacted by mean betweenness centrality in a network (B), and positively by weighted nestedness (C), particularly when species had higher trait variation. In each of these networks, only the species with the highest degree, was positively perturbed from a very low density, $N_i < 0.005$, for a duration of 500 time points with a forcing strength of 0.5, while the rest of the species remained unperturbed. Shown here are data from 115 networks. $\sigma_i$ was fixed at 0.02 for the high trait variation case and 0.005 for the low trait variation case, respectively. Initial mean trait values were sampled according to parameter values given in Table 1. Different colored lines represent generalized linear model fitting with quasibinomial error distributions. Underlying data and R scripts for reproducing this figure can be found in https://doi.org/10.5281/zenodo.13598906.
(TIF)

**S11 Fig. Impact of plant–pollinator weighted nestedness and trait variation on plant–pollinator density after perturbation was stopped.** As nestedness increased proportion of species with density greater than 0.5 increased after species-specific perturbation was stopped and more so when species had high trait variation. To start with, the networks were collated from web-of-life database and species had an initial density of $N_i < 0.005$, and perturbation/forcing strength of 0.5 was applied to the species with the highest degree for a duration of 500 time points. For high trait variation $\sigma_i$ for all species was fixed at 0.02. Initial mean trait values were sampled as given in Table 1. Different colored lines with confidence interval represent generalized linear model fitting with quasibinomial error distributions. Underlying data and R scripts for reproducing this figure can be found in https://doi.org/10.5281/zenodo.13598906.
(TIF)

**S12 Fig. Impact of plant–pollinator network modularity and trait variation on plant–pollinator density after perturbation was stopped.** There was no specific relationship of network modularity and proportion of species with density greater than 0.5 after species-specific perturbation was stopped for 2 levels of trait variation. To start with, the networks were collated from web-of-life database and species had an initial density of $N_i < 0.005$, and perturbation/forcing strength of 0.5 was applied to the species with the highest degree for a duration of 500 time points. For high trait variation $\sigma_i$ for all species was fixed at 0.02. Initial mean trait values were sampled as given in Table 1. Different colored lines with confidence interval represent generalized linear model fitting with quasibinomial error distributions. Underlying data and R scripts for reproducing this figure can be found in https://doi.org/10.5281/zenodo.13598906.
(TIF)

**S13 Fig. Impact of plant–pollinator median network betweenness centrality and trait variation on plant–pollinator density after perturbation was stopped.** At low levels of $\gamma_0 < 1.25$, particularly for high trait variation, low median betweenness centrality would lead to low proportion of species with density greater than 0.5 after species-specific perturbation was stopped. We observe similar results for low trait variation but for $\gamma_0$ of 1.24, 1.3, and 1.35. At levels of $\gamma_0 > 1.2$, networks are already in the recovery parameter range with species-specific perturbation. To start with, the networks were collated from web-of-life database and species had an initial density of $N_i < 0.005$, and perturbation/forcing strength of 0.5 was applied to the species with the highest degree for a duration of 500 time points. For high trait variation $\sigma_i$ for all species was fixed at 0.02. Initial mean trait values were sampled as given in Table 1. Different colored lines with confidence interval represent generalized linear model fitting with quasibinomial error distributions. Underlying data and R scripts for reproducing this figure can be found in https://doi.org/10.5281/zenodo.13598906.
(TIF)

**S14 Fig. Mean of indirect effects per network in relation to recovery biomass of plant–pollinator networks for 2 levels of individual variation and different levels of $\gamma_0$.** (A) Mean net indirect effects, i.e., $\beta$, was below zero across different $\gamma_0$ and for all matrices with spectral radius less than 1. (B) Similarly, mean indirect effects of order 3, $\beta^3$, was also negative across different levels of $\gamma_0$, and decreased slightly as recovery biomass increased. As recovery biomass slightly increased, $\beta^3$, became less negative. Underlying data and R scripts for reproducing this figure can be found in https://doi.org/10.5281/zenodo.13598906.
(TIF)

**S15 Fig. Mean indirect effects faced by a species in a network in relation to mean threshold interaction strength $\gamma_0$.** (A) Mean indirect effects of order 3, $\beta^3$. At low $\gamma_0$, mean indirect effects are mostly negative and remains below 0, but became less negative as $\gamma_0$ increased indicating a positive impact of higher $\gamma_0$. (B) Similarly, net mean indirect effects of all orders, $\beta$, in relation to $\gamma_0$. Net indirect effects are mostly below zero. To be noted that indirect effects shown here fulfills the condition of equation 11 in S1 Text, and it happens when spectral radius of net interactions are less than 1. Underlying data and R scripts for reproducing this figure can be found in https://doi.org/10.5281/zenodo.13598906.
(TIF)

**S16 Fig. Alternative form of mutualistic interaction kernel, the asymmetric kernel as opposed to a Gaussian interaction kernel of the results of the main text.** Left: $\gamma(z, z\prime) = 10\Gamma(((z - z\prime)/W + 10), 4.5, 0.5)$, where an asymmetric function was given as $\Gamma(Z, \alpha_1, \beta_1)$ is the probability density function of a gamma distribution with shape parameter given by $\alpha_1$, and rate parameter given by $\beta_1$. Here $W = 0.1$. Right: the Gaussian interactioin kernel

given in the main text. Underlying data and R scripts for reproducing this figure can be found in https://doi.org/10.5281/zenodo.13598906.
(TIF)

**S17 Fig. Network resurrection from perturbing a single species in a 17 species plant–pollinator network where species interaction was either mediated by Gaussian interaction function as in main text or by an asymmetric interaction kernel (see S16 Fig).** (B) When there is no species specific forcing, at $\gamma_0 = 1.2$, the network remains at the undesirable state. (C) At forcing strength of 0.5 for the species with the highest degree, and with species interaction dictated by a asymmetric kernel, we do not observe network recovery after the perturbation was stopped at 500 time points. (D) However, at $\gamma_0 = 1.3$, we do see a network recovery, but the overall biomass achieved was low. (E) Same network, with Gaussian kernel, without forcing, the network remains at the undesirable state. (F) However, at $\gamma_0 = 1.2$, and species specific forcing for a duration of 500 time points, we see the network recover, while achieving slightly higher biomass than the case of asymmetric species interaction. (G) At higher $\gamma_0$ if 1.3, the same network with Gaussian interaction kernel readily recovers with an overall higher biomass on average. In all these simulations, we considered initial starting species densities to be $N_i < 0.005$ and trait variation for all species to be at a moderate level of $\sigma_i = 0.02$. Underlying data and R scripts for reproducing this figure can be found in https://doi.org/10.5281/zenodo.13598906.
(TIF)

**S18 Fig.** Network recovery from perturbing a single species was impacted positively by positively by nestedness (A), and connectance (B) shown when species interactions were asymmetric in nature, for 5 different thresholds of $\gamma_0$. In each of these networks, only the species with the highest degree, was positively perturbed from a very low density, $N_i < 0.005$, for a duration of 500 time points with a forcing strength of 0.5, while the rest of the species remained unperturbed. Shown here are data from 115 networks. $\sigma_i$ was fixed at 0.02 for the high trait variation case and 0.005 for the low trait variation case respectively. Initial mean trait values were sampled according to parameter values given in Table 1 in main text. In all these simulations, we considered initial starting species densities to be $N_i < 0.005$ and trait variation for all species to be at a moderate level of $\sigma_i = 0.02$. We observe that network recovery is delayed and was achieved at only higher thresholds of mutualistic strength $\gamma_0 > 1.25$. Different colored lines represent generalized linear model fitting with quasibinomial error distributions. Underlying data and R scripts for reproducing this figure can be found in https://doi.org/10.5281/zenodo.13598906.
(TIF)

**S19 Fig. Impact of plant–pollinator network nestedness and trait variation on plant–pollinator density after perturbation was stopped specifically when interaction among species was asymmetric in nature.** (A) As nestedness increased proportion of species with density greater than 0.5 increased after species-specific perturbation was stopped and more so when species had high trait variation, but the threshold at which species achieved densities higher than 0.5 was only at $\gamma_0 > 1.25$, which was not the case when species interacted in Gaussian manner. (B) On average, mean pollinator density across all networks was high when species had high trait variation after recovery for $\gamma_0 > 1$. In (A) to start with, the networks were collated from web-of-life database and species had an initial density of $N_i < 0.005$, and perturbation/ forcing strength of 0.5 was applied to the species with the highest degree for a duration of 500 time points. Different colored lines with confidence interval represent generalized linear model fitting with quasibinomial error distributions. Note that in contrast to Fig 4 in main

text, asymmetric species interaction impacts network recovery. Initial mean trait values were sampled as given in Table 1. Underlying data and R scripts for reproducing this figure can be found in https://doi.org/10.5281/zenodo.13598906.
(TIF)

**S20 Fig. Network resurrection from either random species perturbation or targeted perturbation of 2 mutualistic networks for high trait variation case.** (A, B) The parameter space for successful resurrection for a network with 63 species differed but with a random perturbation of a species for different levels of forcing strength and $\gamma_0$. Only at high threshold $\gamma_0 > 1.3$ we observed complete recovery but at high forcing strength of 0.5. (C, D) For the same network, however, a targeted species-specific perturbation by selecting 2 species that had the highest betweenness centrality score. When perturbing this 2 specific species, we observed signficant differences in network recovery even at low levels of $\gamma_0 < 1.1$. (E, F) For a slightly larger network with 102 species, we observed similar results for random species perturbation and for targeted perturbation (G, H). Targeted perturbation of 2 species based on betweenness centrality scores outperformed random targeting of species in a network. In all these simulations, we considered initial starting species densities to be $N_i < 0.005$ and trait variation for all species to be at a moderate level of $\sigma_i = 0.02$. Underlying data and R scripts for reproducing this figure can be found in https://doi.org/10.5281/zenodo.13598906.
(TIF)

**S21 Fig. Network resurrection from either degree-based species perturbation or targeted perturbation of a 2 mutualistic networks for high trait variation case.** (A–D) The parameter space for successful resurrection for a network with 63 species slightly differed between targeted and degree-based perturbation for different levels of forcing strength and $\gamma_0$. Only at high threshold $\gamma_0 > 1.3$ we observed complete recovery but at high forcing strength of 0.5. (E–H) For a slightly larger network with 102 species, we observed similar results for targeted and degree-based species perturbation. Targeted perturbation of 2 species based on betweenness centrality scores and degree-based perturbation only applied to one single species with the highest degree. In all these simulations, we considered initial starting species densities to be $N_i < 0.005$ and trait variation for all species to be at a moderate level of $\sigma_i = 0.02$. Underlying data and R scripts for reproducing this figure can be found in https://doi.org/10.5281/zenodo.13598906.
(TIF)

**S22 Fig. Impact of plant–pollinator network nestedness and trait variation on plant–pollinator density for 3 different types of perturbation (degree-based, random, and targeted), for different thresholds of mutualistic strength $\gamma_0$ of 0.9, 1, 1.1, 1.2, 1.3, 1.4, and for 2 levels of trait-variation.** Degree-based was based on perturbation that is used in the main text. Random perturbation entailed randomly choosing a species in a network and perturbing the species for 500 time points for a forcing strength of 0.5. Finally, targeted perturbation entailed choosing 2 species based on betweenness centrality measure and perturbing those 2 species for a period of 500 time points with strength of 0.5. As nestedness (NODF) increased proportion of species with density greater than 0.5 increased specifically for targeted and degree-based. To start with, the networks were collated from web-of-life database and species had an initial density of $N_i < 0.005$, and perturbation/forcing strength of 0.5 was applied to the species with the highest degree for a duration of 500 time points. For high trait variation $\sigma_i$ for all species was fixed at 0.02. Initial mean trait values were sampled as given in Table 1. Different colored lines representing generalized linear model fitting with quasibinomial error distributions. Underlying data and R scripts for reproducing this figure can be found in https://doi.org/10.5281/

zenodo.13598906.
(TIF)

**S23 Fig. Impact of plant–pollinator network connectance and trait variation on plant–pollinator density for 3 different types of perturbation (degree-based, random, and targeted), for different thresholds of mutualistic strength $\gamma_0$ of 0.8, 0.9, 1, 1.1, 1.2, 1.3, 1.4, and for 2 levels of trait-variation.** Degree-based was based on perturbation that is used in the main text. Random perturbation entailed randomly choosing a species in a network and perturbing the species for 500 time points for a forcing strength of 0.5. Finally, targeted perturbation entailed choosing 2 species based on betweenness centrality measure and perturbing those 2 species for a period of 500 time points with strength of 0.5. As nestedness (NODF) increased proportion of species with density greater than 0.5 increased specifically for targeted and degree-based. To start with, the networks were collated from web-of-life database and species had an initial density of $N_i < 0.005$, and perturbation/forcing strength of 0.5 was applied to the species with the highest degree for a duration of 500 time points. For high trait variation $\sigma_i$ for all species was fixed at 0.02. Initial mean trait values were sampled as given in Table 1. Different colored lines representing generalized linear model fitting with quasibinomial error distributions. Underlying data and R scripts for reproducing this figure can be found in https://doi.org/10.5281/zenodo.13598906.
(TIF)

**S24 Fig. Impact of variation in Gaussian kernel width, $\omega$, on network recovery from perturbing a single species.** Network recovery was positively impacted by nestedness shown for 2 levels of trait variation (high variation and low variation), for 4 levels of mutualistic strength, $\gamma_0$, of 0.9, 1, 1.1, 1.2, 1.3, 1.4, and for 4 different Gaussian function width $\omega$. In all these networks, only the species with the highest degree, was positively perturbed from a very low density, $N_i < 0.005$, for a duration of 500 time points with a forcing strength of 0.5, while the rest of the species remained unperturbed. Shown here are data from 115 networks. $\sigma_i$ was fixed at 0.02 for the high trait variation case and 0.005 for the low trait variation case, respectively. We observed that higher Gaussian kernel width, $\omega > 0.15$, leads to higher recovery. Initial mean trait values were sampled according to parameter values given in Table 1. Different colored lines representing generalized linear model fitting with quasibinomial error distributions. Underlying data and R scripts for reproducing this figure can be found in https://doi.org/10.5281/zenodo.13598906.
(TIF)

**S25 Fig. Impact of variation in Gaussian kernel width, $\omega$, on network recovery from perturbing a single species.** Network recovery from perturbing a single species was positively impacted by connectance shown for 2 levels of trait variation (high variation and low variation), for 4 levels of threshold mutualistic strength, $\gamma_0$, of 0.9, 1, 1.1, 1.2, 1.3, 1.4, and for 4 different Gaussian function widths $\omega$. In all these networks, only the species with the highest degree was positively perturbed from a very low density, $N_i < 0.005$, for a duration of 500 time points with a forcing strength of 0.5, while the rest of the species remained unperturbed. Shown here are data from 115 networks. $\sigma_i$ was fixed at 0.02 for the high trait variation case and 0.005 for the low trait variation case, respectively. We observed that higher Gaussian kernel width, $\omega > 0.15$, leads to higher recovery. Initial mean trait values were sampled according to parameter values given in Table 1. Different colored lines representing generalized linear model fitting with quasibinomial error distributions. Underlying data and R scripts for reproducing this figure can be found in https://doi.org/10.5281/zenodo.13598906.
(TIF)

**S26 Fig. Impact of variation in heritability on network recovery from perturbing a single species.** Network recovery from perturbing a single species was positively impacted by nestedness shown for 2 levels of trait variation (high variation and low variation), for 4 levels of threshold mutualistic strength, $\gamma_0$, of 0.9, 1, 1.1, and 1.2, 1.3, 1.4, and for 5 different heritability values of 0.1, 0.2, 0.3, 0.4, and 0.5. In all these networks, only the species with the highest degree, was positively perturbed from a very low density, $N_i < 0.005$, for a duration of 500 time points with a forcing strength of 0.5, while the rest of the species remained unperturbed. Shown here are data from 115 networks. $\sigma_i$ was fixed at 0.02 for the high trait variation case and 0.005 for the low trait variation case, respectively. We observed that higher heritability leads to better recovery. Initial mean trait values were sampled according to parameter values given in Table 1. Different colored lines representing generalized linear model fitting with quasibinomial error distributions. Underlying data and R scripts for reproducing this figure can be found in https://doi.org/10.5281/zenodo.13598906.
(TIF)

**S27 Fig. Impact of variation in heritability on network recovery from perturbing a single species.** Network recovery from perturbing a single species was positively impacted by connectance shown for 2 levels of trait variation (high variation and low variation), for 4 levels of threshold mutualistic strength, $\gamma_0$, of 0.9, 1, 1.1, and 1.2, 1.3, 1.4 and for 5 different heritability values of 0.1, 0.2, 0.3, 0.4, and 0.5. In all these networks, only the species with the highest degree was positively perturbed from a very low density, $N_i < 0.005$, for a duration of 500 time points with a forcing strength of 0.5, while the rest of the species remained unperturbed. Shown here are data from 115 networks. $\sigma_i$ was fixed at 0.02 for the high trait variation case and 0.005 for the low trait variation case, respectively. We observed that higher heritability leads to better recovery. Initial mean trait values were sampled according to parameter values given in Table 1. Different colored lines representing generalized linear model fitting with quasibinomial error distributions. Underlying data and R scripts for reproducing this figure can be found in https://doi.org/10.5281/zenodo.13598906.
(TIF)

**S28 Fig. Impact of plant–pollinator network nestedness and trait variation on plant–pollinator density after perturbation was stopped for 2 levels of trait variation (high variation and low variation), 4 levels of threshold mutualistic strength, $\gamma_0$ (columns), 0.9, 1, 1.1, 1.2, 1.3, 1.4, and for 4 different handling time values of 0.1, 0.2, 0.3, 0.4, and 0.5.** As nestedness (NODF) increased proportion of species with density greater than 0.5 increased after species-specific perturbation was stopped and more so when species had high trait variation. To start with, the networks were collated from web-of-life database and species had an initial density of $N_i < 0.005$, and perturbation/forcing strength of 0.5 was applied to the species with the highest degree for a duration of 500 time points. For high trait variation $\sigma_i$ for all species was fixed at 0.02. Initial mean trait values were sampled as given in Table 1. Different colored lines representing generalized linear model fitting with quasibinomial error distributions. Underlying data and R scripts for reproducing this figure can be found in https://doi.org/10.5281/zenodo.13598906.
(TIF)

**S29 Fig. Impact of plant–pollinator network connectance and trait variation on plant–pollinator density after perturbation was stopped for 2 levels of trait variation (high variation and low variation), 4 levels of threshold mutualistic strength, $\gamma_0$ (columns), 0.9, 1, 1.1, 1.2, 1.3, 1.4, and for 5 different handling time values of 0.1, 0.2, 0.3, 0.4, and 0.5.** As connectance increased proportion of species with density greater than 0.5 increased after species-

specific perturbation was stopped and more so when species had high trait variation. To start with, the networks were collated from web-of-life database and species had an initial density of $N_i < 0.005$, and perturbation/forcing strength of 0.5 was applied to the species with the highest degree for a duration of 500 time points. For high trait variation $\sigma_i$ for all species was fixed at 0.02. Initial mean trait values were sampled as given in Table 1. Different colored lines representing generalized linear model fitting with quasibinomial error distributions. Underlying data and R scripts for reproducing this figure can be found in https://doi.org/10.5281/zenodo.13598906.
(TIF)

**S30 Fig. Impact of variation in growth rate, $b_i$, on network recovery from perturbing a single species.** Network recovery from perturbing a single species was positively impacted by nestedness shown for 2 levels of trait variation (high variation and low variation), for 5 levels of threshold mutualistic strength, $\gamma_0$, of 0.9, 1, 1.1, 1.2, 1.3, 1.4, and for 5 different growth rate values of −0.2, −0.1, 0, 0.1, and 0.2. Higher positive growth rate leads to better network recovery. Negative and zero growth rates would indicate that species are obligate mutualists. In all these networks, only the species with the highest degree was positively perturbed from a very low density, $N_i < 0.005$, for a duration of 500 time points with a forcing strength of 0.5, while the rest of the species remained unperturbed. Shown here are data from 115 networks. $\sigma_i$ was fixed at 0.02 for the high trait variation case and 0.005 for the low trait variation case, respectively. Initial mean trait values were sampled according to parameter values given in Table 1. Different colored lines representing generalized linear model fitting with quasibinomial error distributions. Underlying data and R scripts for reproducing this figure can be found in https://doi.org/10.5281/zenodo.13598906.
(TIF)

**S31 Fig. Impact of variation in growth rate, $b_i$, on network recovery from perturbing a single species.** Network recovery from perturbing a single species was positively impacted by connectance shown for 2 levels of trait variation (high variation and low variation), for 5 levels of threshold mutualistic strength, $\gamma_0$, of 0.9, 1, 1.1, 1.2, 1.3, 1.4, and for 5 different growth rate values of −0.2, −0.1, 0, 0.1, and 0.2. Higher positive growth rate leads to better network recovery. Negative growth rates would indicate that species are obligate mutualists. In all these networks, only the species with the highest degree was positively perturbed from a very low density, $N_i < 0.005$, for a duration of 500 time points with a forcing strength of 0.5, while the rest of the species remained unperturbed. Shown here are data from 115 networks. $\sigma_i$ was fixed at 0.02 for the high trait variation case and 0.005 for the low trait variation case, respectively. Initial mean trait values were sampled according to parameter values given in Table 1. Different colored lines representing generalized linear model fitting with quasibinomial error distributions. Underlying data and R scripts for reproducing this figure can be found in https://doi.org/10.5281/zenodo.13598906.
(TIF)

**S32 Fig. Impact of variation in initial trait distributions on network recovery from perturbing a single species.** Network recovery from perturbing a single species was positively impacted by nestedness shown for 2 levels of trait variation (high variation and low variation), for different levels of threshold mutualistic strength, $\gamma_0$, of 0.9, 1, 1.1, 1.2, 1.3, and 1.4 for 2 different initial trait distributions of $U[−0.25, 0.25]$ and $U[−0.75, 0.75]$. Sampling of trait values from a wider trait distribution of $U[−0.75, 0.75]$ leads to poor network recovery from collapse particularly when species have low trait variation. In all these networks, only the species with the highest degree was positively perturbed from a very low density, $N_i < 0.005$, for a duration

of 500 time points with a forcing strength of 0.5, while the rest of the species remained unperturbed. Shown here are data from 115 networks. $\sigma_i$ was fixed at 0.02 for the high trait variation case and 0.005 for the low trait variation case, respectively. Different colored lines representing generalized linear model fitting with quasibinomial error distributions. Underlying data and R scripts for reproducing this figure can be found in https://doi.org/10.5281/zenodo.13598906.
(TIF)

**S33 Fig. Impact of variation in initial trait distributions on network recovery from perturbing a single species.** Network recovery from perturbing a single species was impacted positively impacted by connectance shown for 2 levels of trait variation (high variation and low variation), for different levels of threshold mutualistic strength, $\gamma_0$, of 0.9, 1, 1.1, 1.2, 1.3, and 1.4 for 2 different initial trait distribution values of $U[-0.25, 0.25]$ and $U[-0.75, 0.75]$. Sampling of trait values from a wider trait distribution of $U[-0.75, 0.75]$ leads to poor network recovery from collapse particularly when species have low trait variation. In all these networks, only the species with the highest degree was positively perturbed from a very low density, $N_i < 0.005$, for a duration of 500 time points with a forcing strength of 0.5, while the rest of the species remained unperturbed. Shown here are data from 115 networks. $\sigma_i$ was fixed at 0.02 for the high trait variation case and 0.005 for the low trait variation case, respectively. Different colored lines representing generalized linear model fitting with quasibinomial error distributions. Underlying data and R scripts for reproducing this figure can be found in https://doi.org/10.5281/zenodo.13598906.
(TIF)

**S34 Fig. Network recovery from perturbing a single species was impacted positively impacted by nestedness shown for 2 levels of trait variation (rows: high variation, low variation), for different levels of threshold mutualistic strength, $\gamma_0$, of 0.9, 1, 1.1, 1.2, 1.3, and 1.4 (columns).** In all these networks, only the species with the highest degree was positively perturbed from a very low density, $N_i < 0.005$, for a duration of 500 time points with a forcing strength of 0.5, while the rest of the species remained unperturbed. Shown here are data from a range of 92 artificially generated networks that vary from a low nestedness of 0.28 to a maximum nestedness of 1 while connectance was fixed at 0.42. $\sigma_i$ was fixed at 0.02 for the high trait variation case and 0.005 for the low trait variation case, respectively. Here, the network size was 30 species and connectance was 0.42, and then nestedness matrices keeping the connectance and network size constant were created that varied from low of 0.28 to high of 1. Different colored lines representing generalized linear model fitting with quasibinomial error distributions. Underlying data and R scripts for reproducing this figure can be found in https://doi.org/10.5281/zenodo.13598906.
(TIF)

**S35 Fig. Degree distribution of first 20 plant–pollinator networks.** Data sets and their detailed information could be available in web-of-life database as well as attached in the GitHub repository. Underlying data and R scripts for reproducing this figure can be found in https://doi.org/10.5281/zenodo.13598906.
(TIF)

**S36 Fig. Degree distribution of 20 plant–pollinator networks.** Data sets and their detailed information could be available in web-of-life database as well as attached in the Github repository. Underlying data and R scripts for reproducing this figure can be found in https://doi.org/10.5281/zenodo.13598906.
(TIF)

**S37 Fig. Degree distribution of the 20 plant–pollinator networks.** Data sets and their detailed information could be available in web-of-life database as well as attached in the Github repository. Underlying data and R scripts for reproducing this figure can be found in https://doi.org/10.5281/zenodo.13598906.
(TIF)

**S38 Fig. Degree distribution of the final 30 plant–pollinator networks.** Data sets and their detailed information could be available in web-of-life database as well as attached in the Github repository. Underlying data and R scripts for reproducing this figure can be found in https://doi.org/10.5281/zenodo.13598906.
(TIF)

## Author Contributions

**Conceptualization:** Gaurav Baruah, Meike J. Wittmann.

**Formal analysis:** Gaurav Baruah.

**Funding acquisition:** Gaurav Baruah.

**Investigation:** Gaurav Baruah, Meike J. Wittmann.

**Methodology:** Gaurav Baruah.

**Resources:** Meike J. Wittmann.

**Software:** Gaurav Baruah.

**Supervision:** Meike J. Wittmann.

**Validation:** Gaurav Baruah.

**Visualization:** Gaurav Baruah.

**Writing – original draft:** Gaurav Baruah.

**Writing – review & editing:** Gaurav Baruah, Meike J. Wittmann.

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
