## [Editor Report · Decision Letter 0]

12 Jan 2024

Dear Dr Baruah, 

Thank you for submitting your manuscript entitled "Reviving collapsed ecological networks from a single species" for consideration as a Research Article by PLOS Biology.

Your manuscript has now been evaluated by the PLOS Biology editorial staff, and I'm writing to let you know that we would like to send your submission out for external peer review.

Once your full submission is complete, your paper will undergo a series of checks in preparation for peer review. After your manuscript has passed the checks it will be sent out for review. To provide the metadata for your submission, please Login to Editorial Manager (https://www.editorialmanager.com/pbiology) within two working days, i.e. by Jan 16 2024 11:59PM.

Kind regards,

Roli Roberts

Roland Roberts, PhD

Senior Editor

PLOS Biology

rroberts@plos.org

---

## [Decision Letter · Decision Letter 1]

24 Apr 2024

Dear Dr Baruah,

Thank you for your patience while your manuscript "Reviving collapsed ecological networks from a single species" was peer-reviewed at PLOS Biology. It has now been evaluated by the PLOS Biology editors, an Academic Editor with relevant expertise, and by two independent reviewers. In addition, the Academic Editor has kindly provided some additional guidance (see the foot of this email). I'd just like to extend my further apologies for the extreme delay in processing your manuscript, which is now by far the "oldest" manuscript on our system. This was due to difficulties recruiting reviewers, team absences and problems communicating with the Academic Editor.

You'll see that reviewer #1 says that your study tackles “an important and timely question” using a novel approach. However, s/he has a list of 8 significant criticisms, including a failure to justify your parameter values, and a need to explain how you assessed the quality of the simulations. S/he also thinks that you need to do sensitivity analyses and to study the effects of varying multiple parameters at a time. S/he thinks the model itself is somewhat limited and involves a number of assumptions, etc., etc. Reviewer #2 calls this “a well-written and comprehensive analysis” which “addresses a significant ecological problem with depth and relevance.” However s/he then has a list of 11 points; some of these are trivial, but two involve significant additional analyses (Comment 5, 6); notably there’s some overlap with reviewer #1, with requests for sensitivity analyses and justification for parameter value choice. The Academic Editor emphasises the need to address the reviewers' concerns, and adds a number of his/her own requests.

In light of the reviews and the Academic Editor's comments, which you will find at the end of this email, we would like to invite you to revise the work to thoroughly address the reviewers' reports.

Given the extent of revision needed, we cannot make a decision about publication until we have seen the revised manuscript and your response to the reviewers' comments. Your revised manuscript is likely to be sent for further evaluation by all or a subset of the reviewers.

**IMPORTANT - SUBMITTING YOUR REVISION**

*Re-submission Checklist*

*Published Peer Review*

*PLOS Data Policy*

Sincerely,

Roli Roberts

Roland Roberts, PhD

Senior Editor

PLOS Biology

rroberts@plos.org

REVIEWERS' COMMENTS:

Reviewer #1:

[see attachment for formatted version]

Summary:

The authors investigate how to restore mutualistic ecological networks that have been disrupted by environmental changes. They apply a dynamical eco-evolutionary model informed by empirical plant-pollinator networks and signal propagation theory to examine how network structure, trait variation, and species-specific perturbation influence network recovery. They show that reverting to the original environmental conditions is insufficient for network recovery, but that manipulating a single generalist species can revive the network even in adverse conditions. They also show that network nestedness and trait variation enhance network revival. They suggest that their findings have implications for the conservation and restoration of mutualistic networks.

Review:

The authors address an important and timely question in ecology, namely the resilience and restoration of mutualistic networks. They use a novel combination of theoretical and empirical methods to explore how to restore collapsed networks from a single species. The manuscript is well-written, clear, and concise. The methods are well-explained and the results are supported by figures and tables. The discussion is insightful and connects the findings to the existing literature. The manuscript makes a significant contribution to the field and has the potential to stimulate further research on network dynamics and restoration. However, I have some comments that I think the authors should address before the manuscript can be accepted for publication.

(1) The authors introduce a new model for the problem of interest, but they do not provide any analytical discussion or justification for their approach. They simply set all the parameters according to Table 1 and run numerical simulations. However, they do not explain how they selected the parameter values, nor how they assessed the quality of the simulations. Therefore, the robustness and validity of their results are questionable. The authors should either derive some theoretical analysis or approximation for their model, or conduct more comprehensive and rigorous numerical simulations to demonstrate its performance and limitations under different conditions and assumptions.

(2) The authors do not perform any sensitivity analysis or robustness checks on their model results. They only vary one parameter at a time (the average mutualistic strength, the forcing strength, or the duration of perturbation) and report the outcomes. However, it is possible that the model behavior and the network revival depend on the interactions and combinations of multiple parameters, as well as on the initial conditions and the randomness of the model. It would be useful to explore how the model results change under different scenarios and parameter values, and to test the effects of parameter uncertainty and variability on the model predictions.

(3) The authors adopt a phenomenological model of mutualistic interactions that assumes a Gaussian interaction kernel and a type-2 functional response. While this model has been widely used in previous studies, it has some limitations and assumptions that may affect the generality and applicability of the results. For instance, the model does not account for the effects of spatial structure, dispersal, density-dependence, or frequency-dependence on network dynamics and stability. Furthermore, the model assumes that the phenotypic traits of the species are continuous and normally distributed, which may not be realistic for some plant-pollinator systems. The authors should discuss these limitations and assumptions in more detail and justify their choice of model. They should also investigate the robustness and sensitivity of their results to different model parameters and assumptions, such as the shape and width of the interaction kernel, the handling time, the heritability, and the initial trait distribution.

(4) The authors use a measure of network nestedness (NODF) that is based on the binary adjacency matrix of the network. However, this measure does not account for the variation in interaction strength or frequency among the species, which may also affect the network dynamics and resilience. The authors should consider using a weighted measure of network nestedness that incorporates the interaction strength or frequency. In addition, they should also report and analyze the values of other relevant network metrics, such as modularity, degree distribution, and centrality.

(5) The authors use a perturbation regime that consists of increasing the density of a single species by a constant factor for a fixed duration of time. However, this perturbation regime may not be very realistic or representative of the types of disturbances that occur in natural or human-modified systems. For example, the perturbation may be stochastic, spatially heterogeneous, or affect multiple species simultaneously. The authors should explore the effects of different types of perturbations on network recovery, such as random, targeted, or cascading perturbations, and compare them with their baseline perturbation regime. They should also discuss the ecological and management implications of their perturbation regime and how it relates to real-world scenarios of network restoration.

(6) The authors do not provide a clear definition or operationalization of the concept of network revival which is the first keyword. They seem to use different criteria and thresholds to measure the recovery of the network, such as the proportion of species with density greater than 0.5, the mean pollinator density, the mean plant density, and the mean trait values. However, they do not justify why they chose these criteria and thresholds, and how they relate to the original functionality and diversity of the network. The authors should provide a clear and consistent definition of network revival and explain how they measured it.

(7) The introduction is well-written, but it could be improved by providing more background and context on the problem of network collapse and restoration. The authors should cite more relevant studies on the causes, consequences, and indicators of network collapse, as well as the challenges and strategies of network restoration. They should also state more clearly the main objectives and hypotheses of their study.

(8) The discussion section is insightful, but it could be more balanced and critical. The authors should acknowledge the limitations and uncertainties of their study, and discuss the potential sources of error or bias in their model and analysis. They should also compare and contrast their results with those of other studies, and suggest directions for future research.

Reviewer #2: 

This paper presents a well-written and comprehensive analysis of the revival of collapsed ecological networks, particularly focusing on the intricate dynamics of plant-pollinator interactions. The study addresses a significant ecological problem with depth and relevance, offering valuable insights into strategies for restoring and maintaining the functionality of mutualistic networks in the face of environmental challenges. The authors utilize eco-evolutionary dynamical modeling to investigate the revival of collapsed ecological networks, emphasizing the significance of network architecture and individual variation in managing recovery. Despite efforts to restore original environmental conditions, hysteresis often prevents full recovery, especially in complex mutualistic networks. Instead, the authors propose perturbing species with high interaction numbers, highlighting the importance of trait variation in facilitating network recovery. Overall, focusing on generalist species through positive perturbation emerges as a promising strategy to restore functionality to collapsed networks, even under unfavorable environmental conditions. My comments offer minor suggestions to further refine the quality of the literature.

Comment1: 

In light of the study's focus on positive perturbation strategies targeting generalist species, how do the authors address the potential challenges associated with implementing large perturbations in generalist species compared to smaller perturbations across multiple species, particularly in terms of practical feasibility, ecological risk, and ecosystem resilience? The ecological meaning of forcing/perturbation is not clear in this article. In Figure 2, Figure 3 and other places, the author mentions γ0 = 1.15 + forcing; sometimes it misleads. I suggest clarifying by stating 'With forcing at γ0 = 1.15,' and another case could be 'without forcing.

Comment2:

The title of the paper, "Reviving collapsed ecological networks from a single species," accurately captures the main focus of the study on the revival of collapsed ecological networks using interventions targeting a single species. However, given that the paper specifically discusses plant-pollinator networks and not broader ecological networks, it may be more appropriate to refine the title to better reflect this specificity. A title such as "Reviving collapsed plant-pollinator networks from a single species" would provide readers with a clearer understanding of the specific ecological context under investigation and align more closely with the content of the paper.

Comment3:

How might the effectiveness of positive perturbation strategies vary across different types of ecological networks (e.g., competitive, predator-prey) and under different environmental contexts?

Comment4:

While the author discusses strategies for network revival, it does not address the long-term sustainability or stability of the revived networks. Are there potential risks or challenges associated with the proposed revival strategies, such as the potential for destabilizing feedback loops or unintended consequences for ecosystem functioning?

Comment5:

It is noteworthy that alterations in nestedness can lead to changes in connectance, modularity, and dimension within ecological networks. Consequently, attributing variations in results solely to nestedness becomes challenging. One potential approach to address this concern could involve controlling for dimension and connectance while generating nestedness matrices. By verifying that the results remain consistent under these controlled conditions, it would bolster the confidence in attributing observed changes to nestedness. Alternatively, performing a principal component analysis (PCA) and incorporating nestedness, connectance, and dimension as variables could provide further insights into their individual contributions to the observed outcomes. This approach would enhance the robustness and interpretability of the study findings by identifying the main dimensions of variation and their relationships with the outcomes of interest. Additionally, incorporating the results obtained after generating nestedness matrices or discussing this methodological approach in the literature would strengthen the credibility of the study.

Comment6:

Discussing the robustness of the results for nonzero $b$ values would enhance the applicability and generalizability of the findings. Including a sensitivity analysis or exploring the effects of nonzero $b$ values on network dynamics could provide valuable insights into how variations in species intrinsic growth rates interact with other factors, such as network architecture and perturbation strategies, to influence network resilience and recovery from collapse. 

Comment7:

The discussion on single-species perturbation lacks clarity regarding whether the perturbation targets pollinators, plants, or both. Clarifying the focus of the perturbation, whether it pertains to pollinators, plants, or both, would provide greater insight into its potential effectiveness and utility in network resurrection. Considering the ecological roles and interactions of both pollinators and plants, specifying the targeted species for perturbation could help elucidate which approach may be more beneficial for network recovery. This clarification would enhance the comprehensibility and applicability of the study findings.

Comment8:

The absence of references or empirical data for certain parameter values, such as h^2=0.4, raises concerns about the validity and reliability of the model assumptions. It would greatly enhance the credibility and transparency of the study's findings if supporting evidence or experimental data for these parameter values were provided.

Comment9:

The mention of strategies such as maintaining the abundance of an influential pollinator and setting the decay rate of another influential pollinator to zero is attributed to the paper "Harnessing tipping points in complex ecological networks." Please cite this paper and mention the results. 

Comment10:

The discussion should include future directions or potential applications of the research problem to provide insights for future studies and practical implementations. Incorporating these considerations can enrich the discussion and enhance the relevance of the study's findings.

Comment11:

Correction suggested: Adjust the interval for $u_i$ to [-0.5, 0.5] in line 230.

I also think the authors should make all of their code available, so others can reproduce their work and study

variations. I don't think this has been done.

COMMENTS FROM THE ACADEMIC EDITOR (lightly edited):

The manuscript addresses the problem of restoration of full ecological networks from a theoretical perspective based on signal propagation theory and an eco-evolutionary model based on network structures. The authors explore the idea of hysteresis and how that could define different recovery trajectories during networks rebuilding. Hysteresis itself may work as a brake for functional recovery; this point is original and worth pursuing. Some of the ideas presented are truly innovative, and with potential general interest. Thus I’m positive to offer the possibility for a resubmission.

I’ve myself read the MS in detail and concur with the two referees comments. Overall the MS is well written and structured. Authors should pay attention to the reviewers' suggestions.

Theoretical-based approaches usually demand more realism, yet this also results in much greater model complexity and is often impossible to obtain. However, I think the authors could do a better leverage of their results by providing some assessment of sensitivity analysis and robustness to variations of some key aspects e.g., using a Gaussian kernel to estimate interaction strengths, binary-based nestedness estimates, or the biological bases to set initial parameter values, etc. I mean, I don’t like to be over-demanding in this respect, yet a better discussion and added caution with these key parameters would be welcome.

A missing issue not really addressed in the model approach is the question of indirect effects within networks and how a recovery trajectory may or may not adequately recover such indirect effects. This may be complex, as such indirect effects grow very quickly with increasing number of species and interactions. I don’t mean this would require re-analysis, yet, might be a point worth deserving some discussion. For example, if indirect effects are frequent in these complex networks (any they are, certainly), then this could buffer the responses against hysteresis and somehow increase the likelihood of recovery.

The authors use a dynamical eco-evolutionary framework, to revive mutualistic networks from an undesirable alternative stable state to a high-functioning stable state at unfavorable environmental conditions. The starting point is a single species scenario that may look too simplistic from an empirical, biologically realistic perspective. The idea of setting the focus on a few species whose dynamics could steer the entire network to resurrection is interesting, yet I miss further explanation of how such a distinct subset of species might be a priori identified.

Minor NOTES:

1. In Fig 1, panel A will look too much dense. Try to set thinner line thickness and/or increase a transparency alpha value for the line color. See for instance the adequate line thickness in Fig 3, panels I and L. Also well set in Fig S8 of Suppl Mat.

---

## [Decision Letter · Decision Letter 2]

19 Aug 2024

Dear Dr Baruah,

Thank you for your patience while we considered your revised manuscript "Reviving collapsed plant-pollinator networks from a single species" for publication as a Research Article at PLOS Biology. This revised version of your manuscript has been evaluated by the PLOS Biology editors, the Academic Editor, and the original reviewers.

Based on the reviews, we are likely to accept this manuscript for publication, provided you satisfactorily address the following data and other policy-related requests.

IMPORTANT - please attend to the following:

a) Please put the references into correct PLOS format, with numbered in-line citations.

b) Please address my Data Policy requests below; specifically, we need you to supply the numerical values underlying Figs 1ABCDEF, 2ABCDEF, 1ABCDEFGHIJKLMN, 4ABC, 5AB, S1ABC, S2, S3ABCDEFGHI, S4ABCDEFGHI, S5, S6AB, S7AB, S8ABCDEFGHI, S9ABCDEF, S10ABC, S11, S12, S13, S14AB, S15AB, S16, S17ABCDEFG, S18AB, S19AB, S20ABCDEFGH, S21ABCDEFGH, S22-S38, either as a supplementary data file or as a permanent DOI’d deposition. I note that you already have an associated GitHub deposition (https://github.com/GauravKBaruah/04-Network_revival_git), but this only has the raw network data and the R scripts for the main Figs. Please could you also supply data and/or code required to reproduce the supplementary Figs too, and confirm that the data and code are sufficient to generate the Figs? Also, because Github depositions can be readily changed or deleted, please make a permanent DOI’d copy (e.g. in Zenodo) and provide this URL (see below).

c) Please cite the location of the data clearly in all relevant main and supplementary Figure legends, e.g. “The data underlying this Figure can be found in S1 Data” or “The data underlying this Figure can be found in https://zenodo.org/records/XXXXXXXX

We expect to receive your revised manuscript within two weeks. 

*Published Peer Review History*

*Press*

Sincerely,

Roli Roberts

Roland Roberts, PhD

Senior Editor

rroberts@plos.org

PLOS Biology

DATA POLICY:

Regardless of the method selected, please ensure that you provide the individual numerical values that underlie the summary data displayed in the following figure panels as they are essential for readers to assess your analysis and to reproduce it: Figs 1ABCDEF, 2ABCDEF, 1ABCDEFGHIJKLMN, 4ABC, 5AB, S1ABC, S2, S3ABCDEFGHI, S4ABCDEFGHI, S5, S6AB, S7AB, S8ABCDEFGHI, S9ABCDEF, S10ABC, S11, S12, S13, S14AB, S15AB, S16, S17ABCDEFG, S18AB, S19AB, S20ABCDEFGH, S21ABCDEFGH, S22-S38. NOTE: the numerical data provided should include all replicates AND the way in which the plotted mean and errors were derived (it should not present only the mean/average values).

CODE POLICY

DATA NOT SHOWN?

REVIEWERS' COMMENTS:

Reviewer #1: 

The revised manuscript successfully addresses all my concerns, especially parameter selection, sensitivity analysis as well as ecological and management implications. I recommend the publication of this manuscript.

Reviewer #2:

The author has addressed my questions correctly. I recommend the article for publication if the editor agrees.

---

## [Editor Report · Decision Letter 3]

30 Aug 2024

Dear Dr Baruah,

Thank you for the submission of your revised Research Article "Reviving collapsed plant-pollinator networks from a single species" for publication in PLOS Biology. On behalf of my colleagues and the Academic Editor, Pedro Jordano, I'm pleased to say that we can in principle accept your manuscript for publication, provided you address any remaining formatting and reporting issues. These will be detailed in an email you should receive within 2-3 business days from our colleagues in the journal operations team; no action is required from you until then. Please note that we will not be able to formally accept your manuscript and schedule it for publication until you have completed any requested changes.

Sincerely, 

Roli Roberts

Senior Editor

PLOS Biology

rroberts@plos.org